# TIGIT can inhibit T cell activation via ligation-induced nanoclusters, independent of CD226 co-stimulation

Jonathan D. Worboys [1], Katherine N. Vowell[2], Roseanna K. Hare [1], Ashley R. Ambrose [1], Margherita Bertuzzi [3], Michael A. Conner[2], Florence P. Patel[2], William H. Zammit [1], Judit Gali-Moya[1,4], Khodor S. Hazime [1,4], Katherine L. Jones[1], Camille Rey[1], Stipan Jonjic [5], Tihana Lenac Rovis[5], Gillian M. Tannahill[6], Gabriela Dos Santos Cruz De Matos[6], Jeremy D. Waight[2] & Daniel M. Davis [1,4] ✉

TIGIT is an inhibitory receptor expressed on lymphocytes and can inhibit T cells by preventing CD226 co-stimulation through interactions *in cis* or through competition of shared ligands. Whether TIGIT directly delivers cell-intrinsic inhibitory signals in T cells remains unclear. Here we show, by analysing lymphocytes from matched human tumour and peripheral blood samples, that TIGIT and CD226 co-expression is rare on tumour-infiltrating lymphocytes. Using super-resolution microscopy and other techniques, we demonstrate that ligation with CD155 causes TIGIT to reorganise into dense nanoclusters, which coalesce with T cell receptor (TCR)-rich clusters at immune synapses. Functionally, this reduces cytokine secretion in a manner dependent on TIGIT's intracellular ITT-like signalling motif. Thus, we provide evidence that TIGIT directly inhibits lymphocyte activation, acting independently of CD226, requiring intracellular signalling that is proximal to the TCR. Within the subset of tumours where TIGIT-expressing cells do not commonly co-express CD226, this will likely be the dominant mechanism of action.

Therapeutic targeting of inhibitory receptors on immune cells has led to the development and implementation of an entirely new modality for treating numerous cancers[1]. Despite durable clinical responses to monoclonal antibodies targeting the inhibitory immune receptors CTLA-4 and PD-(L)1, many patients fail to benefit from these approaches, emphasising the need to identify novel and/or complementary therapies. T-cell immunoreceptor with immunoglobulin and ITIM domains (TIGIT) is another inhibitory receptor found on lymphocytes[2–4]. TIGIT binds to multiple nectin and nectin-like ligands CD155, CD112, CD113 and Nectin-4. These ligands are expressed on antigen presenting cells (APC) and often upregulated on cancer cells[4,5]. TIGIT binds with the highest affinity to CD155 and outcompetes the costimulatory receptor CD226 (or DNAM-1) and inhibitory receptor CD96 that bind with lower affinity to the same ligands, akin to the B7/CD28/CTLA-4 pathway[6]. In T cells, TIGIT expression is induced upon activation, again analogous to other inhibitory receptors[2,4,7]. Tumour-infiltrating lymphocytes (TIL) express high levels of TIGIT and blocking antibodies that inhibit the TIGIT-Nectin interaction have demonstrated pre-clinical and clinical tumour control[8–11], facilitating a growing number of clinical trials targeting TIGIT across multiple cancers[12].

[1]Lydia Becker Institute of Immunology and Inflammation, Faculty of Biology, Medicine and Health, Manchester Academic Health Science Centre, University of Manchester, Manchester, UK. [2]GlaxoSmithKline, Collegeville, PA, USA. [3]Manchester Fungal Infection Group, Faculty of Biology, Medicine and Health, Manchester Academic Health Science Centre, University of Manchester, Manchester, UK. [4]Department of Life Sciences, Sir Alexander Fleming Building, Imperial College London, South Kensington London, UK. [5]Center for Proteomics, Faculty of Medicine, University of Rijeka, Rijeka, Croatia. [6]GlaxoSmithKline, Stevenage, UK. ✉e-mail: d.davis@imperial.ac.uk

Multiple mechanisms have been proposed for how TIGIT exerts inhibition in T cells[13]. TIGIT possesses T-cell intrinsic inhibitory potential[7,14–16]. The cytoplasmic domain of TIGIT contains two tyrosine residues, both of which can be phosphorylated in T cells[17], and both within inhibitory motifs[3,18]. Y225 is situated within the immunoglobulin tyrosine tail (ITT)-like motif and Y231 within the immunoreceptor tyrosine-based inhibitory (ITIM) motif. Whilst studies in NK cells have revealed the importance of these motifs[3,18–20], the functional significance of these phosphorylation sites in T cells is currently unknown. TIGIT can also indirectly inhibit T cells by disrupting CD226-mediated co-stimulation[8,17]. FRET-based and immunoprecipitation analysis has determined that CD226 and TIGIT can interact *in cis*, which disrupts CD226 dimer formation and subsequent binding to CD155 on other cells. The cytoplasmic domain of TIGIT is dispensable for indirect inhibition of CD226[17].

Immunoreceptors interact with their ligands at the immune synapse (IS) – the specialised structure that forms at the contacts immune cells make with other cells[21–23]. Distinct nanoscale clusters of receptors assemble at the IS and regulate immune cell activation. TIGIT has been previously shown to accumulate at the IS when it is ligated by CD155 expressed on an interacting cell[17,18]. How this synaptic accumulation of TIGIT relates to T cell activation and signal integration has not yet been explored. In contrast, numerous studies have imaged other inhibitory receptors at the IS, which have helped elucidate their inhibitory mechanism of action[24–28].

In this study we demonstrate that T and NK cells within both the blood and tumours of renal and lung cancer patients seldom co-express TIGIT and CD226, suggesting that within certain tumours TIGIT acts predominantly independently of CD226 *cis* interactions. Using an array of nanoscopic imaging techniques to study the spatiotemporal dynamics of TIGIT in T cells under different activation and ligation conditions, we demonstrate that, upon stimulation, TIGIT organises into nanoscale clusters upon ligation with CD155 that coalesce with the T cell receptor (TCR). By testing the consequences of a variety of point mutations in TIGIT, we found that clustering of TIGIT at the IS is solely dependent on its ability to ligate CD155, but competent ITT-like signalling is required for T cell inhibition. Thus, TIGIT can mediate direct inhibition within T cells, through TCR-proximal inhibitory signalling.

## Results

### TIGIT and CD226 co-expression is infrequent across T and NK cell subsets in renal and lung cancer patient tumours

Recent evidence suggests TIGIT functions by direct inhibition of CD226 *in cis*. Therefore, we evaluated the expression patterns of TIGIT and CD226 in T and NK cells from cancer patients. Matched blood and tumour samples from treatment-naïve patients with clear cell renal cell carcinoma (ccRCC, $n = 8$), nonsmall cell lung cancer (NSCLC, $n = 4$) and lung squamous cell carcinoma (SCC, $n = 2$; Supplementary Table 1) were obtained and studied using a bespoke 40-parameter mass cytometry panel developed to interrogate protein expression of TIGIT and CD226 across immune cell subsets, with a particular focus on T cells (Supplementary Table 2). The subsets within tumours and the blood differed substantially, with <5% of intratumoural T cells and >70% of T cells in the blood being CD45RA+ (Supplementary Fig. 1). To make fair comparisons across tissues, TIGIT expression on CD45RA- T cells (comprising of $T_{CM}$ and $T_{EM}$ subsets) was first investigated. In line with previous reports, the frequency of TIGIT expression was significantly elevated on tumour-infiltrating CD4+ Tregs compared to those in the blood (59.1% ± 7.1% versus 37.8% ± 3.7%), and slightly elevated in the tumour-infiltrating non-Treg CD4 + T cell compartment (16.9% ±3.4% versus 9.1% ± 1.2%; Fig. 1a). The relative expression levels of TIGIT per cell was also increased on CD4+ Tregs and non-Treg CD4 + T cells as determined by median metal intensity (MMI) on TIGIT+ cells (Fig. 1a). Unlike the CD4 + T cell subsets, both the frequency and level of TIGIT

expression did not differ between CD8 + T cells found within the blood or in the tumour (Fig. 1a).

Next, TIGIT and CD226 co-expression was quantified on both CD45RA- T cells and NK cells. Notably, the frequency of CD226 expression on TIGIT + T cells was >2-fold lower in cells from tumours compared to those from blood (Supplementary Fig. 2a). The proportion of intratumoural CD226+ cells within the TIGIT+ populations was most frequent on non-Treg CD4 + T cells (21% ± 2.8%), followed by CD8 + T cells (15.0% ± 3.3%), CD4+ Tregs (11.7% ± 1.4%) and NKs (8.7% ± 3.5%). The overall abundance of CD226 + TIGIT+ co-expressing cells was low (<8% mean frequency) for each T cell subset within the tumour (Fig. 1b, c). The frequency of CD226 co-expression with CD96, another inhibitory receptor from the same signalling axis, was also evaluated. CD96 co-expression with CD226 was also significantly more frequent in cells from the blood than from the tumour (>3-fold mean difference in all subsets) and rare in tumours (Highest mean frequency of 7.3% ±1.4% in non-Treg CD4 + T cells; Supplementary Fig. 2b, c). The infrequency of TIGIT or CD96 co-expression with CD226 is also observed in intratumoural NK cells (<3% in both comparisons; Supplementary Fig. 3a, b).

To identify TIGIT expression patterns on more specific subsets of T cells (including CD45RA + T cells) in the blood and tumours of NSCLC, SCC and ccRCC patients, we used the dimensionality reduction algorithm Uniform Manifold Approximation and Projection (UMAP[29]) and an unsupervised clustering technique for flow and mass cytometry data using self-organizing maps (FlowSOM[30]; Fig. 1d, e and Supplementary Fig. 4). This identified 12 T cell metaclusters with unique phenotypes including 5 CD4 + T cell clusters (c1-4, c6), 1 CD4+ Treg cluster (c5), 1 mixed cluster (c7) and 5 CD8+ T cell clusters (c8-12; Fig. 1e, f, Supplementary Fig. 4 and Supplementary Table 3). The frequency of T cells within metaclusters with well-established phenotypes largely matched expectations between blood and tumour samples. For example, naïve CD4 + T cells (c1) were abundant in blood but rare in tumours, whereas exhausted CD8+ T cells (c11) were abundant in tumours but rare in the blood (Fig. 1g, Supplementary Table 3). The frequency of TIGIT expression was highest on CD4+ Tregs (c5, 63.7% ± 4.02%), followed by exhausted CD8+ T cells (c11, 32.3% ± 3.6%), Tfh-like CD4 + T cells (c4, 25.2% ± 3.2%), and a subset of CD8 + T cells with high CD57 expression (c8, 27.4% ± 2.5%) (Fig. 1g, gray bars). The greatest co-expression of TIGIT and CD226 was observed within a small portion of c8, which was more abundant in the blood than the tumour (Fig. 1d; dashed line on top row). In tumours, co-expression was limited to a small subset of c4 (Fig. 1d; dashed line on bottom row). CD226 and CD96 co-expression patterns are much more similar than CD226 and TIGIT, with both c2 and c9 (CD4 Tcm and CD8 TD) demonstrating significant similarity in their UMAP profiles. We also performed the same type of analysis on NK cells (Supplementary Fig. 3c–g, Supplementary Fig. 5, and Supplementary Table 4). TIGIT was expressed on less than 20% of NK cells across all samples, showing little expression variation between tumour and blood-derived cells and was highest on tumour-infiltrating NK cells with a terminally differentiated phenotype (c6; Supplementary Fig. 3g).

CD155 is the dominant ligand for TIGIT and CD226 and can be expressed on myeloid populations. Thus, we next investigated the abundance of CD155 within the same 14 tumour samples plus an additional 5 ccRCC patients, 17 NSCLC patients and 7 lung squamous cell carcinoma (SCC) patients (Supplementary Table 1). Whilst our CD45 sample barcoding strategy prevented us from interrogating CD155 expression on cancer cells, we did observe that between 10 and 80% of myeloid antigen-presenting cells (mAPC) were CD155+. We found moderate positive correlation between the frequency of CD155 expression on mAPC populations and TIGIT and CD96 expression on T and NK cell subsets within the tumours, but not with CD226 or PD-1 (Fig. 1h and Supplementary Fig. 3h). These correlations were specific to the receptor-ligand interactors as no correlations were found between the frequency of CD155 expression on mAPCs and the inhibitory

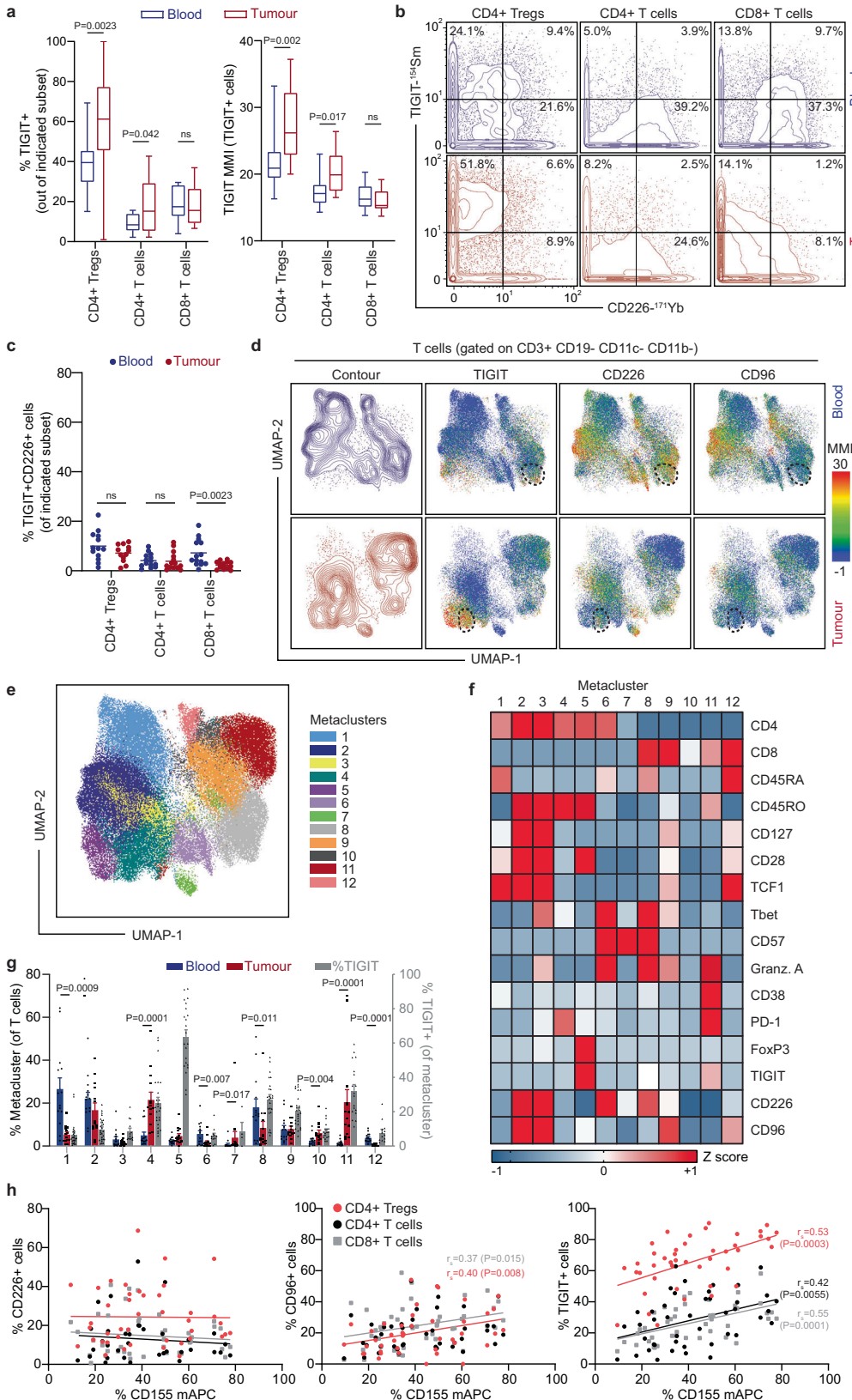

receptor PD-1 on the same T cell subsets (Fig. 1h and Supplementary Fig. 2d).

Taken together, these data suggest TIGIT and CD226 co-expression is rare on immune cells within the tumour microenvironment and that CD155 is abundant on mAPCs for TIGIT-expressing T and NK cells to engage.

## TIGIT accumulates at the Immune Synapse (IS) in a ligand-dependent manner and correlates with inhibition

To test the functional consequences of TIGIT expression in T cells, we established a model cell system whereby receptor-ligand interactions could be controlled. The Jurkat-Raji model was suitable as i) Jurkat-Raji co-cultures have been successfully employed to study T cell activation

**Fig. 1 | Mass cytometry analysis of peripheral blood and dissociated tumour cell (DTC) samples from renal and lung cancer patients shows limited co-expression of CD226 and TIGIT on T cells. a** Frequency (left) and expression level (right) of TIGIT on CD45RA- CD4+ Tregs, CD4 + T cells, or CD8 + T cells in blood (blue) or tumour (red) samples. Relative TIGIT expression intensity was determined by median metal intensity (MMI) of TIGIT+ cells within each subset. Boxes depict the 25th-75th percentile with a line showing the median. Whiskers display minimum to maximum values. $n = 14$ matched patient samples. **b** Co-expression of TIGIT and CD226 was investigated on CD45RA- CD4+ Tregs, CD4+ T cells (non-Tregs), and CD8+ T cells from the blood and tumour. Plots display the frequency of singly or coexpressing populations on concatenated files of the matched blood or tumour samples. **c** Mean frequency of cells co-expressing TIGIT and CD226 in each CD45RA- T cell subset from blood and tumour. $n = 14$ matched samples (parent populations with <50 events excluded). **d** UMAP analysis performed on T cells from matched blood and tumour samples ($n = 14$ of each). Samples with greater than 3,000 T cell events ($n = 19$) were downsampled to 3,000 events prior to concatenation. UMAP projections show concatenated T cells from the blood (top; $n = 35,691$ events) or tumour (bottom; $n = 42,000$ events), highlighting contour (left) or median metal intensity (MMI) of TIGIT, CD226, and CD96 (right). Arbitrary

dashed lines are used to highlight visual examples of discrete and co-expression patterns of TIGIT and CD226. **e** FlowSOM metaclusters were created on T cells concatenated from all matched samples and projected onto the same UMAP as in **d**. **f** Expression intensity heatmap of the indicated markers for each of the 12 FlowSOM metaclusters in **e**. Color scale indicates row-adjusted z-score expression for each marker. **g** Mean frequency (±SEM) of T cells in each metacluster for each individual matched sample ($n = 14$). Gray bars represent the frequency of TIGIT+ events (per biaxial gating) within each metacluster (blood and tumour samples are combined; $n = 28$, except where <50 events in parent population; statistical differences between blood and tumour are listed in Supplementary Table 3). **h** Plots displaying the correlation of the frequency of CD155 expression on myeloid antigen-presenting cells (mAPC) with the frequency of CD226+ (left), CD96+ (middle), and TIGIT+ (right) cells from CD4+ Tregs (red circles), CD4+ T cells (black circles), or CD8+ T cells (gray squares) in the tumour microenvironment ($n = 44$). Linear regression lines and Spearman's Rho ($r_s$) are shown for correlations that were significant (Two-tailed; $P < 0.05$, as indicated). Non-parametric matched-pairs two-tailed Wilcoxon tests were used to determine differences between PBMCs and DTCs. Source data are provided as a Source Data file.

---

in vitro[27], ii) Jurkat cells do not express TIGIT[17] and iii) Raji B cells do not express TIGIT ligands CD155 and CD112[31,32]. Thus, Jurkat cells were transduced to express TIGIT, fused either to eGFP (Jurkat TIGIT-GFP) or SNAP-tag (Jurkat TIGIT-SNAP) at its C-terminus, and Raji cells transduced to express binding (CD155) or non-binding (CD111) nectin ligands, each fused to a C-terminal V5 epitope tag (Fig. 2a, b). Confocal microscopy of conjugates revealed that TIGIT on Jurkat cells did not accumulate at the IS unless the target cell expressed CD155, whereby there was a 4 ± 1-fold increase in abundance at the IS (log2 fold change of 2 ± 0.4; Fig. 2c, d). The use of an antagonistic TIGIT antibody, known to prevent CD155 and TIGIT from interacting, but not an isotype-matched control, prevented this synaptic accumulation (Fig. 2c, d). In human peripheral blood CD4+ and CD8 + T cells (stimulated to induce TIGIT expression; Supplementary Fig. 6), TIGIT accumulated at immune synapses to a similar degree as in Jurkat TIGIT-GFP cells (Fig. 2e, f). Additionally, we employed a reductionist approach whereby we replaced target cells with silica bead-supported lipid bilayers (BSLB; Supplementary Fig. 7a). BSLBs could be loaded with specific densities of ligands and conjugated to Jurkat TIGIT-GFP cells. These results phenocopied our observations with cell-cell conjugates, demonstrating that artificial lipid bilayer systems can faithfully reproduce cellular interactions, and showing that accumulation of TIGIT at the IS does not require any additional components present in a *bone fide* target cell (Supplementary Fig. 7b, c).

It is still unclear whether TIGIT is directly inhibitory in T cells. To test the functional contribution of TIGIT expressed in Jurkat cells, stimulation with Raji cells pulsed with the superantigen Staphylococcal Enterotoxin E (SEE) was used (Fig. 2g). Parental Jurkat cells, co-cultured for 6 h with SEE-pulsed Raji-CD155 cells, released more IL-2 than when co-cultured with SEE-pulsed Raji-CD111 cells (Log2-fold increase 0.18 ± 0.2; Fig. 2h; Supplementary Fig. 8a). This increase is likely due to the low levels of CD226 expressed on Jurkat cells (Supplementary Fig. 8b). As parental cells do not express TIGIT, the increased IL-2 release with Raji-CD155 cells does not change in the presence of an antagonistic TIGIT antibody, nor its isotype control. However, when TIGIT-SNAP was expressed in Jurkat cells, co-culture with Raji-CD155 now led to ~37% as much IL-2 release when compared with Raji-CD111 co-cultures (Log2-fold decrease of 1.45 ± 0.2). This reduction in IL-2 release with Raji-CD155 co-cultures could be almost completely rescued with an antagonistic TIGIT antibody (Log2-fold decrease of 0.09 ± 0.2) but not with an isotype matched control (Log2 fold decrease of 1.54 ± 0.2). Thus, TIGIT accumulates at the IS when interacting cells express its ligand CD155 leading to direct inhibition of T cell activation.

## TIGIT accumulates in clusters at the IS upon ligation

Receptor-ligand interactions at cell-cell contacts result in specific spatial arrangements that are important in their function[33]. To visualise the organisation of TIGIT and CD155 at the synapse between Jurkat and Raji cells, super-resolution 3D-TauSTED was performed (Supplementary Fig. 9 & Supplementary Video 1). As observed with confocal imaging (Fig. 2c), most of the TIGIT signal is observed at the IS, whereas the CD155 signal is readily visualised throughout the Raji cell surface (Supplementary Fig. 9a, b). At the IS, both the TIGIT and CD155 colocalise in distinct clusters (Supplementary Fig. 9c).

To further evaluate the synaptic organisation of TIGIT, planar lipid bilayers (PLB) containing protein ligands were used to stimulate cells and imaged with TIRF microscopy (Fig. 3a). Jurkat TIGIT-GFP cells were incubated on PLBs containing the integrin ICAM-1, to facilitate cell adhesion, and either CD111 or CD155. Upon interaction with PLBs containing either ICAM-1 alone or with CD111, the TIGIT intensity was weak and diffuse across the synapse (Fig. 3b). The addition of CD155 into the PLBs resulted in greater TIGIT intensity at the synapse, with TIGIT accumulating into multiple dense clusters. Such reorganisation was abrogated with an antagonistic TIGIT antibody. The extent and consistency of the observed TIGIT clustering was quantified as the fraction of pixels that displayed fluorescence at 1.5 times that of the mean pixel intensity within each cell (Clustering index; Fig. 3c).

Primary human CD4+ and CD8+ T cells were assessed similarly, through interaction with PLBs containing either ICAM-1 with CD111 or CD155. In both cell types, TIGIT accumulated at the interface upon ligation with CD155 (Fig. 3d). The synaptic distribution of TIGIT in primary CD4 + T cells was less punctate than seen in Jurkat cells, yet we observed a similar increase in the degree of clustering as with Jurkat TIGIT-GFP cells (Fig. 3e). In CD8 + T cells, TIGIT was more centrally located and despite observing clear intensity increases, the clustering index was unable to quantify differences between ligation conditions. This could highlight a difference in the spatial organisation of TIGIT between specific T cell subsets.

Together, these data show that ligation with CD155 either through cell conjugation or interaction with model PLBs brings TIGIT to the IS within defined clusters.

## TIGIT clusters are dynamic and display properties of liquid-liquid phase separation

To visualise the dynamics of TIGIT clusters, TIGIT-SNAP was selectively labelled with a fluorescent dye in live Jurkat cells that can be imaged over long periods with minimal photobleaching[34]. Labelled Jurkat

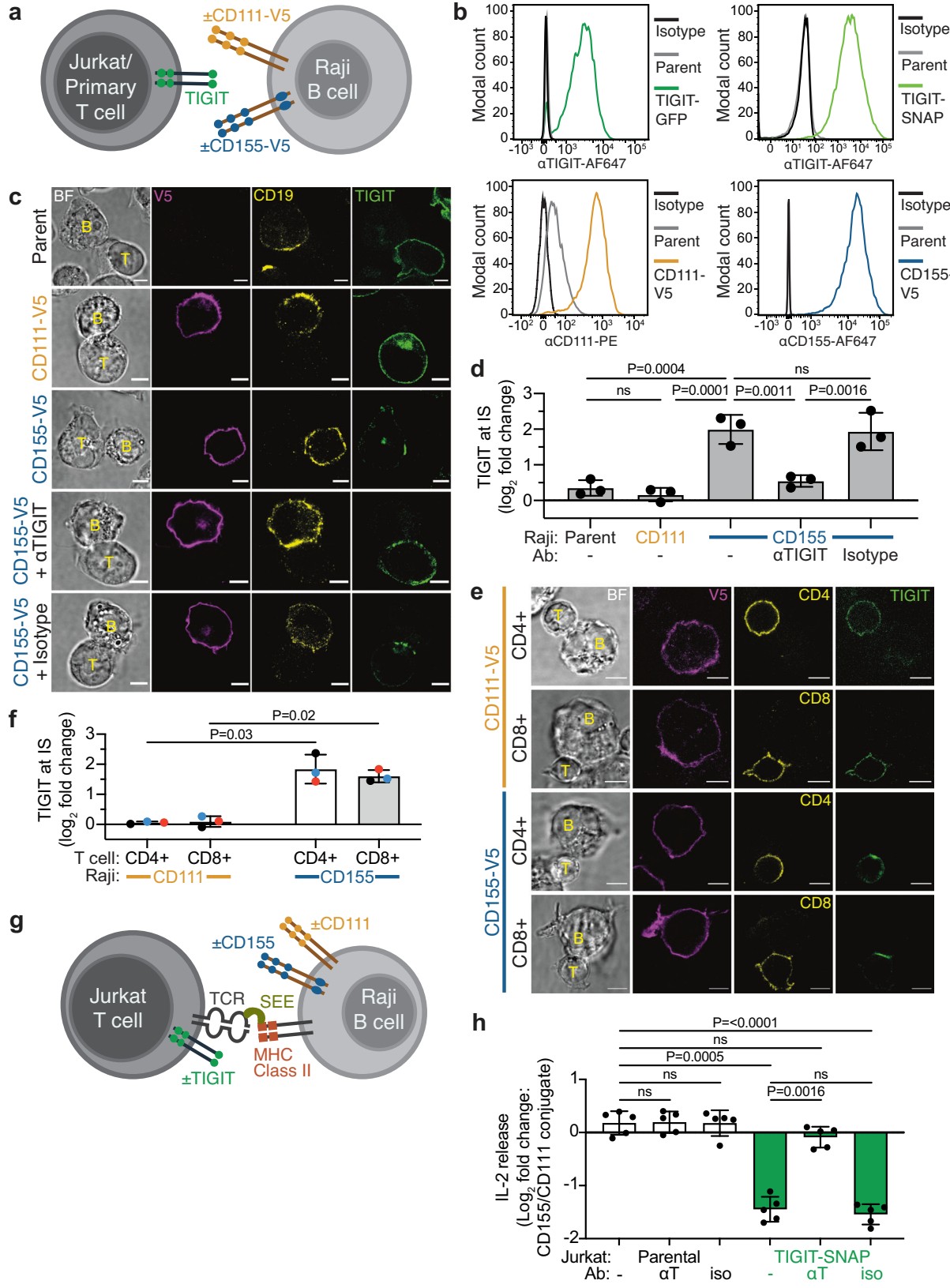

TIGIT-SNAP cells were added to PLBs loaded with ICAM-1 and either CD111 or CD155 and imaged every 3 s (Supplementary Video 2 & Fig. 3f–g). On PLBs presenting CD111, TIGIT remained diffuse on the T cell surface membrane, with some areas of higher intensity that did not show much mobility. On PLBs presenting CD155, the TIGIT signal accumulated in small puncta as soon as it becomes visible in the TIRF plane, indicative of rapid cluster formation, which move centripetally over the 30 minute timeframe. The TIGIT clusters demonstrate lateral mobility and appear to both fuse and separate with other clusters (Fig. 3h).

The visual properties of the cluster dynamics are reminiscent of those observed for molecules that have undergone liquid-liquid

**Fig. 2 | TIGIT accumulates at the Immune Synapse (IS) in a ligand-dependent manner and inhibits T-cell intrinsic activation. a** Schematic depicting the model system employed to visualise TIGIT on the surface of T cells when interacting with Raji B cells expressing different nectin ligands. **b** Flow cytometry analysis showing the expression of TIGIT in Jurkat cells (above) and CD111 and CD155 in Raji cells (below), in both the parental and expression lines together with isotype-matched controls. **c** Confocal microscopy images showing TIGIT-GFP (green) on the surface of Jurkat cells (T) conjugated for 20 mins with different Raji cell (B) populations, as indicated to the left of the panel. CD19 (yellow) is used to mark Raji cells and a V5 stain labels expressed nectins (magenta). Respective brightfield images (BF) are also provided. The bottom two rows show Jurkat T cells that have been pre-incubated with either an antagonistic TIGIT antibody or an isotype-matched control. **d** Mean log$_2$ fold change in synaptic TIGIT enrichment in Jurkat cells, from the conjugates shown in **c** (±S.D.; $n = 3$ independent experiments; adjusted $P$ values from a one-way ANOVA with Tukey's multiple comparisons are given; ns = not significant). **e** Representative confocal microscopy images showing TIGIT (green)

on the surface of primary T cells conjugated with different Raji B cell populations, as indicated to the left. CD4 and CD8 (yellow) were stained to mark T cell subsets, and BF provided. **f** Mean log$_2$ fold change (±S.D., $n = 3$ independent donors matched by colour) in synaptic TIGIT enrichment in primary T cells, from the conjugates shown in **e**. Adjusted $P$ values from a paired T-test are given (Holm-Šídák method). **g** Schematic depicting the model system employed to test the inhibitory effect of TIGIT on the surface of Jurkat T cells when interacting with cells expressing different nectin ligands. Staphylococcal Enterotoxin E (SEE) was used to stimulate Jurkat cells. **h** Relative amounts of IL-2 released from either parental or TIGIT-SNAP-expressing Jurkat cells after co-incubation with SEE-pulsed Raji cells for 6 h. Data is shown as the mean log$_2$ fold changes between Raji-CD155 conjugates compared to Raji-CD111 conjugates (±S.D., $n = 5$ independent experiments with adjusted $P$ values from a one-way ANOVA with Holm-Šídák's multiple comparisons displayed). Cells pre-incubated with an antagonistic TIGIT antibody (αT) or an isotype-matched control (iso) are shown, as indicated. All scale bars = 5 μm. Source data are provided as a Source Data file.

phase separation. Molecules will often be exchanged dynamically between different phase states, and this is thought to be important in how phase separation can create hubs of signalling activity[35,36]. To test whether TIGIT nanoclusters are comprised of static or dynamically exchanging molecules, fluorescence recovery after photobleaching (FRAP) was used. By using CD155 conjugated to a fluorescent dye (AF647) in PLBs, both the ligand and receptor could be photobleached simultaneously and the fluorescence recovery recorded. Both CD155-AF647 in PLBs and TIGIT-GFP recovered after photobleaching within clusters (Fig. 3i), but the positioning of the clusters persisted. CD155 recovered much more rapidly and to a greater total extent (Fig. 3j; Half times: CD155 = 14.85 s, TIGIT = 114.8 s; Total recovery at 300 s: CD155 = 96%, TIGIT = 79%), but this likely reflects the fact that the PLB is not impeded by other proteins or complex underlying structures. Thus, both TIGIT and its ligand are dynamically exchanged in-and-out of fixed-position clusters at the immune synapse.

## TIGIT organises into denser, larger nanoscopic-sized clusters upon CD155 ligation

Direct stochastic optical reconstruction microscopy (dSTORM), which improves the resolution of diffraction limited microscopy by ~10-fold[37], can simultaneously illuminate nanoscale structures not visible at normal resolution and provide 2D localisation maps permitting detailed spatial analysis[38]. Jurkat cells expressing TIGIT-GFP were plated onto PLBs containing ICAM-1, CD111 or CD155, fixed and stained with anti-GFP nanobodies conjugated to Alexa Fluor-647, and imaged with dSTORM using TIRF illumination (Fig. 4a). The increased resolution revealed that TIGIT showed both nanoscale clustering in unligated conditions and dense, submicron-scale clusters upon CD155 ligation (Fig. 4a and Supplementary Fig. 10a). Quantification of the number of localisations at the IS in each condition was determined as a proxy for the number of TIGIT molecules (Fig. 4b). This showed that the density of TIGIT localisations at the IS increased ~3–5-fold with CD155 ligation (with mean values of 22 ± 9, 31 ± 19 & 95 ± 61/μm$^2$ for ICAM-1, CD111 & CD155, respectively). Moreover, TIGIT accumulation was abrogated with an antagonistic TIGIT antibody. Regions (5 × 5 μm) within the IS of these cells were subjected to quantitative Ripley's-based clustering analysis[39] to assess the spatial distribution of the individual events. The Ripley's H function ($L(r) − r$) can demonstrate the extent of clustering, by analysing the number of localisations within increasing concentric radii centred on each localisation and can distinguish random distributions from those that are dispersed or clustered[40]. Plotting the H function against increasing radii can inform on these distributions, with values > 0 indicating clustered distributions. In all conditions measured, TIGIT localisations are clustered (Fig. 4c). The H function peaks at a radius of ~40 nm for un-ligated TIGIT and 360–430 nm when

TIGIT is ligated. This establishes that TIGIT clusters have radii up to ten times greater upon ligation. Further comparison of TIGIT organisation utilised Getis and Franklin's local point pattern analysis to quantitatively define clusters (density and binary maps; Fig. 4a). Quantification of the binary cluster maps show that ligated TIGIT clusters are ~2.5 fold greater in area than non-ligated clusters (Fig. 4d; 12,051 ± 1116, 12,847 ± 1877 & 31,480 ± 23,679 nm$^2$ for ICAM-1, CD111 & CD155, respectively). Additionally, ligation increased the density of molecules within nanoclusters approximately 2-fold (Fig. 4e; 430 ± 48, 418 ± 98 & 779 ± 277/μm$^2$ for ICAM-1, CD111 & CD155, respectively). Overall, the number of localisations within clusters increased upon ligation (Supplementary Fig. 10b), although the number of clusters remained the same (Supplementary Fig. 10c). Small, but statistically significant, differences were observed in the fraction of events deemed to be within clusters, but overall, most localisations were clustered (Supplementary Fig. 10d).

In primary human T cells that expressed TIGIT endogenously, TIGIT constitutively appeared within nanoclusters on both unligated CD4+ and CD8+ T cells, with clusters appearing larger in CD8+ cells (Fig. 4f–g & Supplementary Fig. 10e). Upon ligation with CD155, clusters of TIGIT become larger and denser. The density of localisations is much greater in primary cells than in Jurkat TIGIT-GFP cells, with ~10 times more localisations in the unligated (CD111) control conditions in both subsets (Fig. 4h; 31 ± 19, 319 ± 222 & 327 ± 395 localisations/μm$^2$ for Jurkat TIGIT-GFP, CD4+ and CD8+ , respectively). This likely reflects a greater overall abundance of the receptor on the primary cells. The density of synaptic TIGIT in CD4+ and CD8+ cells increased upon ligation 3.5–5-fold (1,605 ± 801 & 1189 ± 958 localisations/μm$^2$ for CD4+ and CD8+, respectively). Ripley's H functions show clustered distributions in all conditions, with CD155 ligation increasing the radii peaks (Fig. 4l; 40 vs 180 nm in CD4+ and 110 vs 210 nm in CD8+). This higher maximal peak in unligated CD8+ cells is also reflected in the greater mean cluster area compared to unligated CD4+ cells (Fig. 4j; 30,131 ± 13,966 vs 19,285 ± 5,192 nm$^2$, albeit not statistically significant). CD155 ligation created both larger and denser clusters in both subsets (Fig. 4j, k; 31,484 ± 13,607 nm$^2$ in CD4+ & 49,326 ± 26,851 in CD8+; 1611 ± 300 to 2202 ± 842 localisations/μm$^2$ in CD4+ & 1235 ± 549 to 2435 ± 1138 localisations/μm$^2$ in CD8+). Ligation with CD155 caused more TIGIT molecules to be within clusters (Supplementary Fig. 9f) but did not have much impact on the total number of clusters in CD8+ T cells (Supplementary Fig. 9g), analogous to what was seen with Jurkat TIGIT-GFP. In CD4+ T cells, however, the number of clusters significantly decreased upon ligation, which may reflect the smaller nanoclusters observed in CD4+ T cells in unligated conditions (61.3 ± 18.6 vs 39.1 ± 19.4 in CD111 and CD155, respectively; Supplementary Fig. 10g and Fig. 4f). Surprisingly, we observed a dramatic decrease in the fraction of events in clusters in ligated conditions in

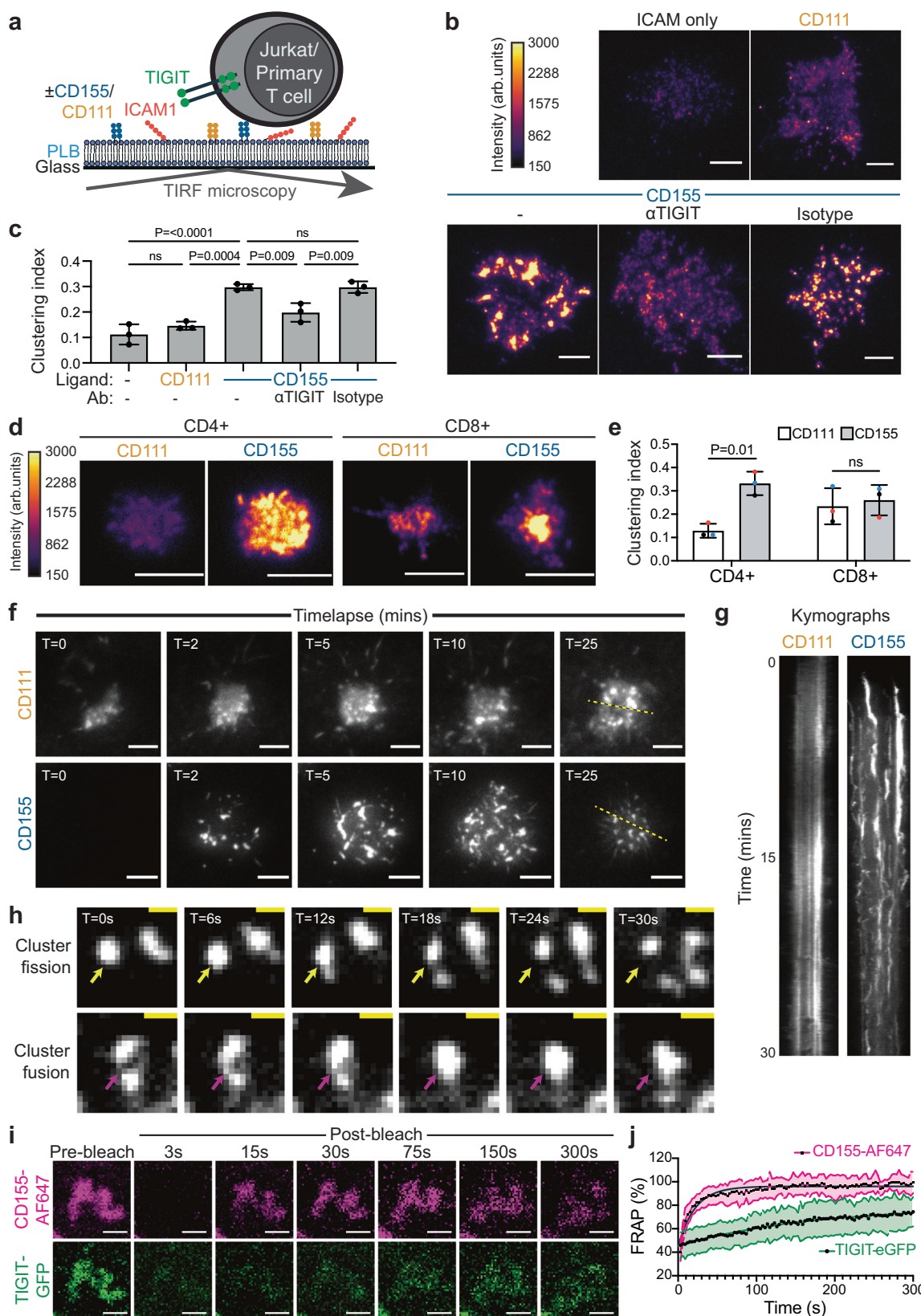

both subsets (79.8 ± 13.3% to 29.3 ± 24.2% in CD4+ and 69.5 ± 24.7% to 48.3 ± 22.9% in CD8+; Supplementary Fig. 10h). Overall, dSTORM analysis demonstrated CD155 ligation caused TIGIT to cluster within submicron-scaled clusters that were between 1.6–2.5 fold larger and 2–5 fold denser than unligated nanoclusters across Jurkat and primary T cells.

## CD155-ligated TIGIT clusters coalesce with ligated TCR

The nanoscale proximity of receptors directly relates to their mechanism of action, whereby signals can either synergise or compete[25,41–43]. Thus, we next sought to investigate the spatial proximity of TIGIT clusters relative to the TCR, upon activation of Jurkat TIGIT-SNAP cells. To test this, the stimulatory TCR antibody OKT3 was

**Fig. 3 | TIGIT assembles into dense, dynamic clusters at the Immune Synapse (IS) in a ligand-dependant manner. a** Schematic depicting the model system employed to visualise TIGIT at the IS of T cells upon ligation. TIGIT expressing T cells interact with Planar Lipid Bilayers (PLB) containing laterally mobile ligands and imaged with Total Internal Reflection Fluorescence (TIRF) microscopy. **b** TIRF microscopy images showing TIGIT-GFP at the IS of Jurkat cells that have interacted with PLBs loaded with ICAM-1 (100 molecules/$\mu m^2$), and either CD111 or CD155 (400 molecules/$\mu m^2$) for 20 mins. Cells preincubated with an antagonistic TIGIT antibody or an isotype-matched control are shown, as indicated. **c** Mean degree of TIGIT clustering measured from the images shown in **b** (±S.D.; $n = 3$ independent experiments with adjusted $P$ values from a one-way ANOVA with Tukey's multiple comparisons shown; ns = not significant). **d** Representative TIRF microscopy images showing the spatial distribution of TIGIT at the IS of primary CD4+ and CD8 + T cells that have interacted with PLBs loaded with ICAM-1, and the ligands CD111 or CD155 for 20 mins, as in **b**. In both **b** and **d** the fluorescent intensities have been scaled equally, and the colour scales provided. **e** Mean degree of TIGIT clustering measured from the images shown in **d** (±S.D., $n = 3$ individual donors).

Adjusted $P$ values from a paired T-test with Holm-Šídák's multiple corrections are displayed. **f** Video stills from live TIRF microscopy imaging of Jurkat T cells expressing TIGIT-SNAP interacting with PLBs containing ICAM-1 and either CD111 or CD155 (as in **b**). Acquisition times are indicated at the top left (mins). **g** Kymographs showing a single spatial position, as indicated by the dashed yellow line in **f**, over time. **h** Zoomed video stills from Jurkat TIGIT-SNAP on PLBs containing ICAM-1 and CD155, from **f**, displaying occurrences where TIGIT clusters appear to split (top row, yellow arrow) or fuse (bottom row, magenta arrow). Arrows mark specific xy locations, and time intervals are displayed above. **i** Confocal microscopy images of a FRAP experiment showing the recovery of both CD155-AF647 within the PLB and TIGIT-GFP on the surface of Jurkat cells within clusters. PLBs contain both ICAM-1 and CD155-AF647. Images were taken before photobleaching (Pre-bleach), and at the indicated times (in seconds) following photobleaching (Post-bleach). **j** FRAP profiles of both CD155-AF647 and TIGIT-GFP from cells measured as shown in **i**. Data is presented as the mean ±S.D. ($n = 11$ cells from 2 independent experiments). Scale bars = 5 $\mu m$ (**b**, **d**, **f**) and 1 $\mu m$ (**h**, **i**). Source data are provided as a Source Data file.

---

added to PLBs in combination with either ICAM-1 and CD111 or ICAM-1 and CD155 (Fig. 5a). OKT3 in PLBs was labelled with a fluorescent dye and TIGIT-SNAP via a SNAP-label and imaged simultaneously (Fig. 5b, c). On stimulatory PLBs, containing CD111, TIGIT remains relatively diffuse across the IS with some degree of overlap with the position of OKT3 (marking the TCR). When CD155 is present in the PLB, the subsequent TIGIT clusters that form strongly colocalise with TCR clusters. Upon ligation, TIGIT clusters moves centripetally with the TCR towards the central supramolecular activation complex (cSMAC; Fig. 5c). In primary human CD4+ and CD8 + T cells, TIGIT ligation also caused TIGIT to co-cluster with the TCR, both at early timepoints when nascent clusters form (3 mins), and within the cSMAC at later timepoints (10 mins; Fig. 5d).

TIGIT shares binding of CD155 with the closely related molecules CD226 and CD96, with CD226 being co-stimulatory and CD96 thought to be inhibitory. We questioned whether ligation of these similar proteins caused the same relative changes in their proximities to the TCR. Thus, 5-day stimulated CD4+ primary T cells were added to PLBs containing ICAM-1 and OKT3, and CD111 or CD155 (Supplementary Fig. 11) and fixed after 10 min of interaction. CD226 did not colocalise with the TCR in unligated conditions, but colocalised to the cSMAC when CD155 was ligated (Supplementary Fig. 11a). CD96 can bind to both CD111 and CD155, and consistent with that CD96 staining was more intense when either CD111 or CD155 was present in the PLB (Supplementary Fig. 11b). Strikingly, however, ligation with either molecule did not bring CD96 and the TCR into proximity, with CD96 visually excluded from TCR clusters.

As receptors interact at nanoscale proximities, we next sought to image TIGIT and the TCR with 2-colour dSTORM. Jurkat TIGIT-GFP, labelled with a GFP nanobody (coupled to Atto-488), and OKT3-AF647 in the PLB was imaged sequentially at early and late timepoints (Fig. 6a, Supplementary Fig. 12a). At both early timepoints where the TCR clusters form and later timepoints where the TCR clusters concentrate at the cSMAC, TIGIT colocalised with the TCR upon ligation at nanoscale proximities. Colocalisation of dSTORM images can be quantified with multiple metrics. Firstly, Spearman's Rank and Mander's correlations of individual channel computed tessellations[44] showed increased correlations between TIGIT and the TCR upon ligation (Spearman's: 0.27 ± 0.09 and 0.61 ± 0.10 and Mander's: 0.21 ± 0.21 and 0.59 ± 0.14 for CD111 and CD155 at 2 min, respectively; Fig. 6b, c and Supplementary Fig. 12b–d). Importantly, blocking the TIGIT-CD155 interaction dramatically reduced this correlation (0.28 ± 0.15 and 0.14 ± 0.13 Spearman's and Mander's correlations for CD155 + anti-TIGIT at 2 min, respectively). As a positive control, we labelled TIGIT with both a GFP nanobody and a TIGIT-directed antibody (Spearman's: 0.73 ± 0.05, Mander's: 0.8 ± 0.11). The XY coordinates of individual localisations were then reversed to generate

randomised localisations to act as negative controls containing the same number of event localisations (Spearman's: 0.24 ± 0.15, Mander's: 0.23 ± 0.13). Secondly, coordinate-based colocalisation (CBC[45]) was used to compare relative proximities of TIGIT and the TCR. CBC scores of >0.8 are considered to be strongly colocalised, and thus the total fraction >0.8 was used to quantify the extent of colocalisation (Fig. 6d and Supplementary Fig. 12e). A greater fraction of localisations was colocalised with TIGIT ligation (15.3 ± 2.9% vs 9.6 ± 2.2% at 2 min and 17.5 ± 2.9% vs 8.7 ± 2.4% at 10 min), with antibody blockade reducing this at both timepoints (9.6 ± 1.8% and 11.0 ± 3.0% at 2 and 10 min, respectively). Positive controls colocalised 32.1 ± 3.8% of localisations whereas the respective negative control colocalised 6.4 ± 1.1%. Lastly, we also measured the proximity between TIGIT and TCR localisations with nearest neighbour distance (NND), which computes the distance of the nearest localisation between the two datasets. Positive controls reached NND values close to the theoretical resolution limit of 20 nm, whereas the negative control peaks at 140 nm (Supplementary Fig. 12f). All conditions where CD155 can ligate TIGIT (dashed lines) gave NND values close to 20 nm, whereas unligated or antibody blocked TIGIT gave values between 80–120 nm.

Next, we investigated whether TIGIT and the TCR were co-proximal upon ligation in primary T cells. As primary T cells are smaller than Jurkat cells, they take longer to sink to the bottom of the imaging well to interact with the PLB. Thus, both CD4+ and CD8 + T cells from peripheral blood were fixed at a slightly later 'early' timepoint (3 minutes instead of 2). Nevertheless, we observed a similar co-proximity of TIGIT and TCR clusters at both early nascent forming clusters and later within the cSMAC (Fig. 6e). Interestingly, we saw that in CD8 + T cells TIGIT and TCR clusters colocalised to a greater extent than in CD4 + T cells. Colocalisation analysis with either the tessellation-based approach or CBC led to increases in correlation upon ligation at both early and late timepoints (Fig. 6f–h). Highest correlations were recorded at 10 min of interaction with the PLB in ligated conditions in both subsets and was highest for CD8+ (Spearman's: 0.66 ± 0.01, 0.55 ± 0.09; Mander's: 0.77 ± 0.06, 0.60 ± 0.07 for CD8+ and CD4+, respectively). NND analysis also demonstrated highest co-proximity in ligated conditions at 10 min (30 nm for CD8+ and 50 nm for CD4+; Supplementary Fig. 12g). All ligated conditions produced smaller NND values than their respective unligated controls. Taken together, TIGIT and the TCR co-cluster at nanoscale proximities upon co-ligation, providing the spatiotemporal context for TIGIT-mediated inhibition.

## TIGIT and CD226 coalesce upon co-ligation yet CD226 does not alter TIGIT clustering

As both TIGIT and CD226 localise to the TCR upon ligation, we hypothesised that these two receptors themselves colocalise. To test this, we incubated activated CD4+ and CD8 + T cells with PLBs

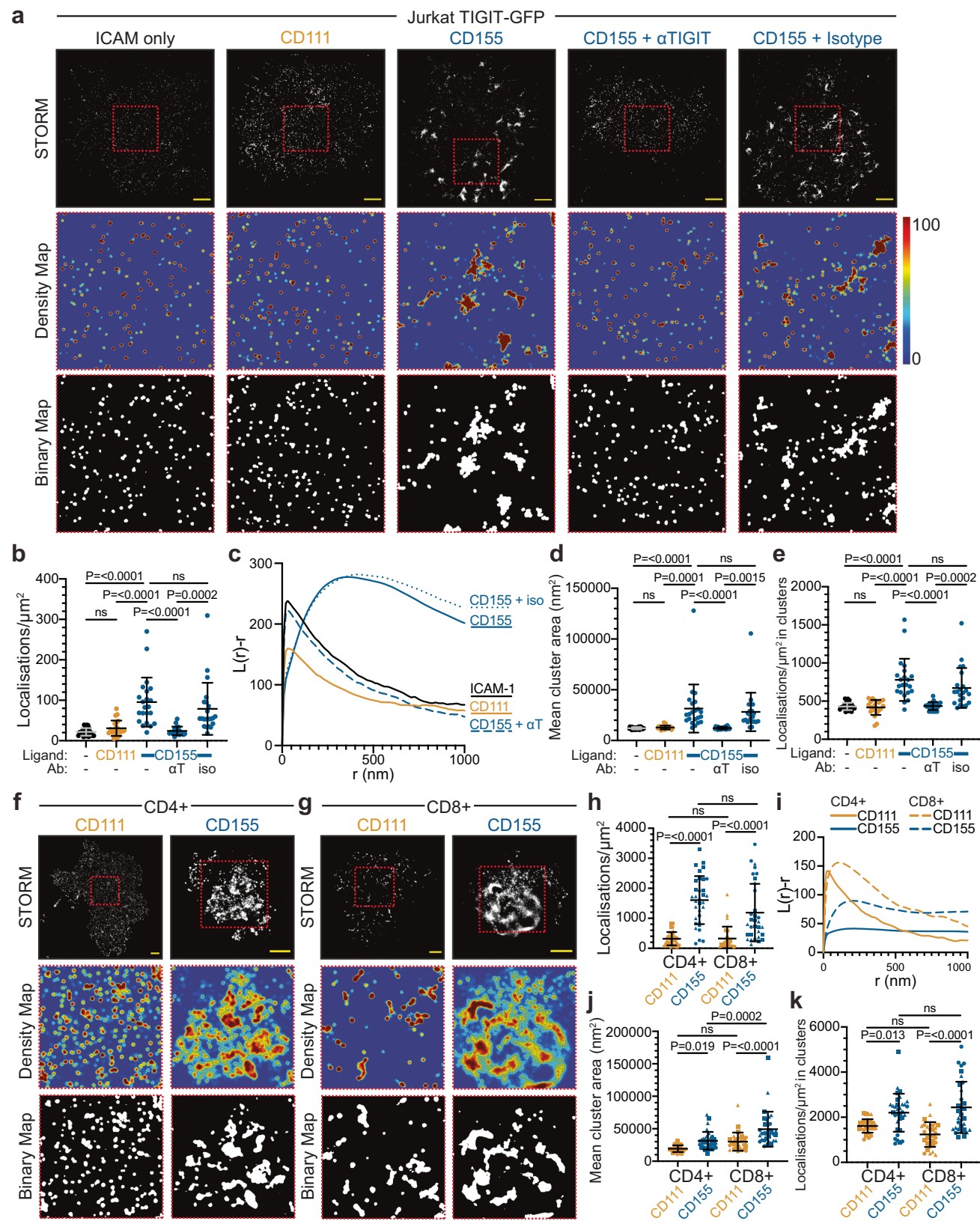

**a**  — Jurkat TIGIT-GFP —

ICAM only | CD111 | CD155 | CD155 + αTIGIT | CD155 + Isotype

STORM

Density Map

Binary Map

**b** Localisations/µm²
P=<0.0001
P=<0.0001  P=0.0002
ns  P=<0.0001  ns
Ligand: - CD111 CD155
Ab: - - - αT iso

**c** L(r)-r
CD155 + iso
CD155
ICAM-1
CD111
CD155 + αT
r (nm)

**d** Mean cluster area (nm²)
P=<0.0001  ns
P=0.0001
ns  P=<0.0001  P=0.0015
Ligand: - CD111 CD155
Ab: - - - αT iso

**e** Localisations/µm² in clusters
P=<0.0001  ns
P=<0.0001  P=0.0002
ns  P=<0.0001
Ligand: - CD111 CD155
Ab: - - - αT iso

**f** —CD4+—
CD111 | CD155
STORM
Density Map
Binary Map

**g** —CD8+—
CD111 | CD155
STORM
Density Map
Binary Map

**h** Localisations/µm²
ns
ns
P=<0.0001  P=<0.0001
CD4+  CD8+
CD111 CD155  CD111 CD155

**i** L(r)-r
CD4+  CD8+
CD111 ---- CD111
CD155 ---- CD155
r (nm)

**j** Mean cluster area (nm²)
ns
P=0.0002
P=0.019  ns  P=<0.0001
CD4+  CD8+
CD111 CD155  CD111 CD155

**k** Localisations/µm² in clusters
ns  ns
P=0.013  P=<0.0001
CD4+  CD8+
CD111 CD155  CD111 CD155

containing ICAM-1, OKT3 and either CD111 or CD155 and performed 2-colour dSTORM analysis of TIGIT and CD226 (Supplementary Fig. 13a). TIGIT and CD226 showed little colocalisation consitutively but coalesced within dense clusters upon ligation by CD155 (Supplementary Fig. 13a–d). As both receptors cluster at the same location, we questioned whether the presence of CD226 impact the clustering of

TIGIT. To test this, we evaluated TIGIT clustering in both TIGIT⁺CD226⁻ and TIGIT⁺CD226⁺ subsets of stimulated primary T cells upon interaction with PLBs containing ICAM-1, OKT3 and CD155 (Supplementary Fig. 14). TIGIT⁺CD226⁻ and TIGIT⁺CD226⁺ subsets were delineated by antibody staining (Supplementary Fig. 14a, b), with TIGIT expression not significantly different between subsets (Supplementary Fig. 14c).

**Fig. 4 | Single molecule localisation microscopy reveals extent of TIGIT clustering upon ligation. a** dSTORM imaging of TIGIT-GFP in Jurkat cells that have interacted with Planar Lipid Bilayers (PLBs) loaded with ICAM-1 (100 molecules/$\mu m^2$), and the indicated nectin ligands (400 molecules/$\mu m^2$) for 20 mins. Cells pre-incubated with an antagonistic TIGIT antibody ($\alpha$T) or an isotype-matched control (iso) are shown, as indicated. Representative STORM images are shown in the top row, scale bar = 2 $\mu m$. Smaller regions (5 $\mu m$ x 5 $\mu m$; dashed red boxes) were subjected to cluster analysis (see methods) and Getis and Franklin density maps and binary maps showing identified clusters are displayed below. **b–e** Quantitative analysis of the single molecule localisation images shown in **a**. n = ≥22 (representing a 25 $\mu m^2$ region from a single cells), examined over 3 independent experiments, with adjusted P values from one-way ANOVA with Tukey's multiple comparisons

shown (ns = not significant). Mean density of TIGIT localisations (**b**), mean Ripley's H function at different clustering radii (**c**), mean cluster area (**d**; with each datapoint representing the mean cluster size per region of a single cell) and mean density of events within clusters (**e**). **f–g** dSTORM imaging of TIGIT in primary peripheral blood-isolated CD4+ (**f**) and CD8+ (**g**) T cells on PLBs loaded with ICAM-1, and the indicated nectin ligands, as in **a**. Representative STORM images are shown in the top row, scale bar = 1 $\mu m$. Smaller regions (3 $\mu m$ x 3 $\mu m$; dashed red boxes) were subjected to cluster analysis, as in **a**. **h–k** Quantitative analysis of the single molecule localisations depicted in **f** and **g**, as in **b–e**. n = ≥30 cells from 3 independent donors, as depicted through symbol shape. Adjusted P values from a one-way ANOVA with Tukey's multiple comparisons are shown. Error bars represent standard deviation throughout. Source data are provided as a Source Data file.

TIGIT clustering between each subset was not significantly different when assessed by TIRF microscopy (Supplementary Fig. 14d) or through various single molecule clustering metrics (Supplementary Fig. 14e–h). Overall, both TIGIT and CD226 can colocalise upon ligation if co-expressed and the presence of CD226 does not impact TIGIT clustering.

### Ligand binding in the absence of downstream signalling is sufficient for TIGIT accumulation and clustering at IS

As TIGIT accumulation at the synapse correlates with its ability to prevent cytokine secretions, we next sought to investigate which functional domains of TIGIT control synaptic accumulation. Thus, we introduced 12 different mutant forms of TIGIT, into the TIGIT-SNAP construct, that affected its glycosylation (N32Q and N101Q), dimerization (I42D), ligand binding (Q56R, N70R and Y113R), downstream signalling (Y225A, N227Q, Y231A and YAYA), as well as introducing a polymorphism with an unknown function (I33V; Fig. 7a). These mutations were introduced based on previous work characterising these residues through structural and genetic studies[18,46,47]. Mutants were generated by site-directed mutagenesis and expressed in Jurkat cells, with flow cytometry and Western blotting confirming similar expression levels, and the N-linked glycosylation pattern previously observed[47] (Supplementary Fig. 15a, b). Confocal microscopy of mutant TIGIT-expressing Jurkat lines conjugated to nectin-expressing Raji lines demonstrated that only the mutations known to disrupt ligand binding (Q56R, N70R and Y113R in the (V/I)(S/T)Q, AX$_6$G, and T(F/Y)P motifs, respectively) caused significant reductions in TIGIT accumulation at the synapse (Fig. 7b, c). Y113R reduced the extent of TIGIT accumulation at the IS, whereas both Q56R and N70R completely abrogated TIGIT accumulation, which is consistent with binding affinities previously reported[46]. A truncated form of TIGIT (T164*) that completely lacks the cytoplasmic tail, and thereby any capacity to signal, accumulated at the synapse akin to wild type (WT) TIGIT, demonstrating that only the extracellular and transmembrane portion is required.

Even though accumulation at the synapse occurred with most mutant forms, we next questioned if this accumulation still occurred in a clustered manner, as this is likely to affect its function. Accordingly, mutant TIGIT-expressing Jurkats were added to PLBs loaded with ICAM-1 and CD155, with the WT TIGIT line also added to bilayers containing ICAM-1 and CD111 as a negative control. Only the Q56R and N70R mutations, that abrogate ligand binding, prevented TIGIT from clustering at the IS upon ligation (Fig. 7d, e). Y113R did show a significant reduction in its ability to cluster, as compared to the WT, consistent with the observations between cell conjugates. This establishes that TIGIT assembly into nanoscale clusters at the IS upon ligation is strictly controlled by binding its ligand CD155 and isn't affected by known post-translational modifications or intracellular signalling.

### T-cell mediated TIGIT inhibition requires ITT-like domain

It is currently not known how any of the mutations we introduced into TIGIT affects its ability to inhibit T cells, although the

intracellular domain is not required to disrupt CD226 signalling[17]. Thus, we next sought to investigate the inhibitory potential of each of the mutant forms in Jurkat-Raji SEE co-cultures. As before, we cultured each of the mutant TIGIT-expressing Jurkats with SEE-pulsed Raji cells, expressing either CD111 or CD155, and measured the amount of secreted IL-2 after 6 h (Fig. 7f). As expected, mutants of TIGIT with diminished binding to CD155 all reduced inhibition in Raji-CD155 conjugates and in a manner that correlated with their binding potential. None of the mutants that affected glycosylation sites, homodimerisation nor the I33V polymorphism were able to significantly affected TIGIT-mediated inhibition, although N32Q did reduce inhibition slightly. However, deletion of the cytoplasmic domain (T164*) and mutation of both tyrosine phosphorylation sites (YAYA) led to increases in IL-2 release compared to cells not expressing TIGIT (albeit only significantly so for T164*). Mutations within the ITT-like domain (Y225A and N227Q), but not the ITIM (Y231A), significantly reduced TIGIT inhibition. To test the relative contribution of Y225 and Y231 to TIGIT phosphorylation we used Phos-tag SDS-PAGE[48] (Fig. 7g). In Jurkat cells expressing WT TIGIT we observed a significant mobility shift, representing the phosphorylated form of TIGIT, following ligation with SEE-pulsed Raji CD155 cells. This phosphorylated form was absent in Jurkat cells expressing both the Y225A and YAYA mutant, and present (as a more mobile band) in cells expressing the Y231A mutant. This indicates that Y225 is the major site of phosphorylation, correlating with it being vital for the inhibitory function of TIGIT in Jurkat T cells.

Additionally, Western blot analysis was performed to analyse the dynamic phosphorylation events in parental, WT TIGIT- and YAYA TIGIT-expressing Jurkat cells following stimulation with SEE-pulsed Raji CD155 cells (Supplementary Fig. 16). We observed the expected dynamic regulation of CD3$\zeta$, Zap70, LAT, ERK1/2, AKT and I$\kappa$B$\alpha$ in parental cells upon activation, yet observed no significant differences in phosphorylation of the proteins in the WT TIGIT expressing line (Supplementary Fig. 16a, b). We did observe a small reduction in pI$\kappa$B$\alpha$ (S32) that was not significant but did not observe any changes to its total abundance. Thus, the signalling cascade downstream of TIGIT in Jurkat cells remains unclear.

As we observed that TIGIT mutants that prevented inhibition could still cluster upon ligation, we then looked to see whether such mutant forms could still co-cluster with the TCR using PLBs (Fig. 7h). As negative controls, we imaged cells expressing WT TIGIT on PLBs containing ICAM-1, OKT3 and CD111, and cells expressing the Q56R mutation on PLBs containing ICAM-1, OKT3 and CD155. In each of these cases, TIGIT remained diffuse at the synapse and correlated poorly with OKT3 using a Pearson's analysis (0.40 ± 0.13 for WT on CD111 and 0.34 ± 0.10 for Q56R on CD155; Fig. 7i). However, all mutants affecting the cytoplasmic signalling domains co-clustered with the TCR, with Pearson's correlations between 0.64 and 0.7 (±0.10–0.14), as compared to 0.69 ± 0.11 for WT (Fig. 7i).

Taken together, these results demonstrate that TIGIT inhibition is biphasic, with clustering proximal to the TCR being dependent on

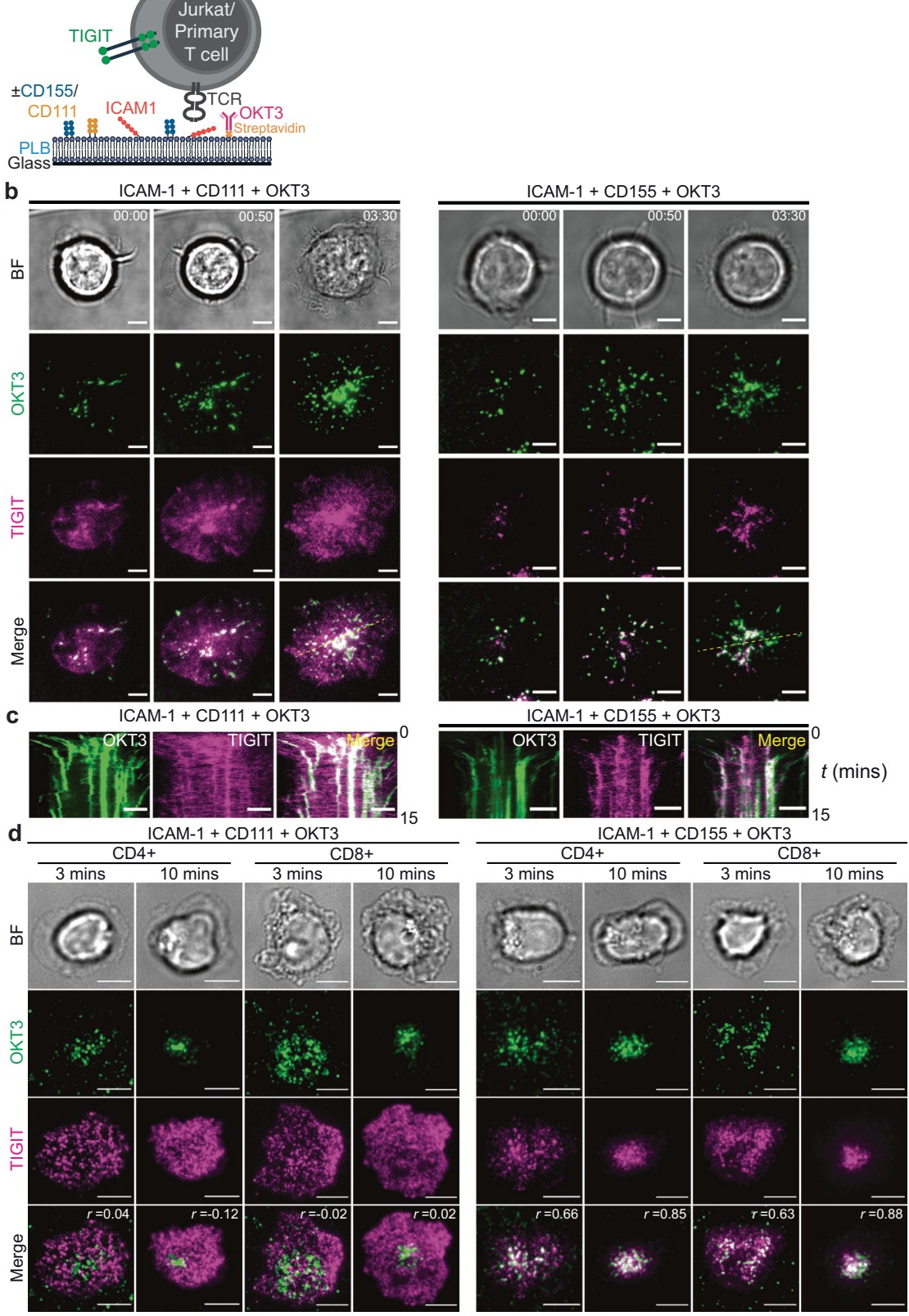

ligand binding and functional inhibition being dependent on cytoplasmic inhibitory domain-dependent signalling.

## Discussion

Here, we show that TIGIT and CD226 are infrequently co-expressed on both peripheral blood and particularly so on tumour infiltrating lymphocytes in renal and lung cancer. This finding is significant as recent work has demonstrated that TIGIT and PD-1 converged to inhibit CD226 signalling and this mechanism was proposed to explain the success of dual blockade of TIGIT and PD-1[17]. In this model, TIGIT blocked CD226 signalling, solely through its extracellular domain, by both competition for its ligand CD155 and through disruption of

**Fig. 5 | TIGIT clusters are proximal to T-cell receptor (TCR) clusters. a** Schematic depicting the model system employed to visualise TIGIT and the TCR at the Immune Synapse (IS) of T cells upon co-ligation. Both Jurkat T cells expressing TIGIT-SNAP, and peripheral blood-isolated primary T cells that express TIGIT endogenously interact with PLBs containing nectin ligands (CD111 or CD155), ICAM-1 and the directly labelled, mono-biotinylated stimulatory TCR antibody OKT3 and imaged with TIRF microscopy. **b** Video stills of Jurkat T cells expressing TIGIT-SNAP and labelled with dye (magenta) interacting with PLBs containing ICAM-1 (100 molecules/$\mu m^2$), CD111 or CD155 (400 molecules/$\mu m^2$) and fluorescently labelled OKT3 (100 molecules/$\mu m^2$; green), using live TIRF microscopy. Acquisition times are indicated at the top right of each column of images (mins:secs). Brightfield images are shown above. The data are representative of 3 independent experiments. **c** Kymographs showing a single spatial position, as indicated by the dashed yellow line in **b**, over time. **d** Representative TIRF microscopy images showing the relative localisation of TIGIT (antibody labelled; magenta) and the TCR (green) upon interaction with PLBs, as in **b**, in fixed primary CD4+ and CD8 + T cells at the indicated times. Throughout, scale bars = 5 μm. The data are representative of 3 independent donors. Pearson correlation coefficients (*r*) are displayed on merged images.

CD226 homodimerisation through direct interactions *in cis*. The data we present would suggest that this model of TIGIT inhibition is likely to contribute to a small fraction of the immune cells in particular cancers. This may not be the case for all cancers, as TIGIT and CD226 were co-expressed in glioblastoma multiforme and in melanoma[49,50]. In the case of co-expression, TIGIT and CD226 may indeed compete to signal, as has been shown in Treg cells[16], and may occur in the vicinity of the TCR. Where TIGIT is singly expressed, our data imply that TIGIT signalling itself, is important for its inhibitory function and would include T-cell intrinsic inhibition, TIGIT-mediated CD155 signalling on APCs[4], and cell extrinsic inhibition through reduced ligand availability for CD226 signalling. This hypothesis is further supported by the correlations we observe between TIGIT+ T cells and CD155+ mAPCs within tumours. Indeed, even in CD226-deficient mice, TIGIT blockade was still able to dramatically reduce tumour burden in a murine cancer model, although mice were in general less able to control tumours[17]. This has important therapeutic implications, as TIGIT blockade could differ mechanistically from PD-1 blockade, and a deeper understanding of how TIGIT and PD-1 synergise might be useful to predict such therapeutic success.

We do not provide evidence of *cis* TIGIT-CD226 in our imaging assays of primary T cells (Supplementary Fig. 13). It is possible *cis* interactions are transient, or that *cis* interactions block the epitope of either antibody we used for immunostaining. Thus, whilst we did not observe TIGIT-CD226 *cis* interactions, we cannot rule out the possibility that they occur.

Given the general downregulation of CD226 expression in the tumour microenvironment due to the presence of immunosuppressive factors, such as TGF-β and hypoxia[51,52] our data showing that TIGIT can inhibit TCR signalling independently of CD226 widens the potential value in application of TIGIT blockade as an immunotherapeutic strategy, compared to a model in which co-expression is required. In addition, we provide greater context for TIGIT expression in the tumour microenvironment which may facilitate a deeper understanding of the mechanistic consequences of blockade. In the T cell compartment, for example, we found that TIGIT was most highly upregulated on CD4+ Tregs as well as CD8 + T cells which co-expressed PD-1, CD38, and Granzyme A. Thus, blockade of TIGIT may serve a dual role in this compartment, reducing Treg immunosuppressive capacity[53] whilst re-invigorating exhausted CD8 + T cells to enhance the antitumour immune response. In contrast, CD226 expression in these compartments was only observed on a small subset of CD127+ TCF-1+ stemlike precursors. Upon TIGIT blockade, this population may be boosted by costimulatory signals that are more likely to occur through CD226-CD155 interactions.

We sought to investigate the spatiotemporal dynamics of the TIGIT-CD155 interaction, with a focus on the membrane proximal events on the surface of the T cells. We discovered that the TIGIT-CD155 interaction leads to an accumulation of the receptors at the IS, within dynamic nanoscale clusters. Clustering of TIGIT was solely dependent on ligand binding and did not require cytoplasmic interactions. How the TIGIT-CD155 interaction creates nanoscopic segregation of the molecules at the surface is yet to be determined, but could be related to topological membrane architecture, or specific molecular interactions (be it protein-, carbohydrate- or lipid-based). It is important to note that most clustering data in this manuscript is generated using supported lipid bilayers, that differ from the cell surface of an APC. Thus, there may be effects from other aspects of APCs, such as the sub-synaptic actin, or specific nanoscale organisation of target cell ligands, which are not captured in our experiments. Additionally, we observed properties of the clusters that are observed in molecules that have undergone liquid-liquid phase separation but whether TIGIT is truly phase separated will need further investigation. TIGIT clustering is likely to be important for its function, as with other membrane receptors, such as the TCR, B-Cell receptor (BCR) and nephrin[54–56]. Intriguingly, we observed clustering in both the YAYA and T164* mutants that lack inhibitory signalling capacity, and a concomitant increase in IL-2 release from these mutant-expressing Jurkats in SEE-Raji conjugates. Therefore, it is possible that the clustering formed upon TIGIT-CD155 interaction can be stimulatory, possibly through force generation or increased adhesion when no inhibitory signals are present. This has important implications for possible chimeric receptors used on CAR T cells or CAR NK cells, or multibinding engagers, being developed as therapies. However, as all mutants that could engage CD155 were able to cluster, we could not directly discriminate the role of clustering for TIGIT function.

As previously seen with PD-1 and CD28, TIGIT clusters co-localised to within nanometres of TCR clusters upon activation. The underlying mechanism driving their proximity will need to be elucidated but may arise from shared interactions. The proximity to the TCR suggests a possible role for TIGIT in directly modulating TCR-signalling, although we observed no reduction in the phosphorylation of CD3ζ, Zap70 or LAT. It is highly likely that TIGIT clustering is important for its signal transduction, as interacting molecules within biomolecular condensates exhibit longer dwell times that give rise to enhanced enzymatic activity[57,58]. In vitro phosphorylation assays show that TIGIT required twice as much Fyn to achieve equivalent phosphorylation levels to PD-1[17]. Clustering may compensate for this by increasing local kinase phosphorylation efficiencies. The same principal is likely to apply to downstream processes. In vitro assays failed to show phosphorylated TIGIT binding to Grb2, SHIP-1 or Shp-1[17], although these interactions have been seen in both T and NK cells[15,18,20]. Perhaps the clustering mechanisms in place within cells provide the threshold needed to produce such interactions.

We provide the first demonstration in T cells that the inhibitory potential of TIGIT relies more on the ITT-like domain than the ITIM, although both played some role. This is consistent with i) the higher levels of phosphorylation observed on Y225 than Y231 in both T and NK cells[17,18], and ii) the greater importance of the ITT-like domain for human NK cell-cytotoxicity[18]. In NK cells, phosphorylation of the ITT-like domain, but less so the ITIM, has been shown to mediate interaction with Grb2 and β-arrestin 2[18,20]. β-arrestin 2 is expressed in T cells[59], and could feasibly play an active role in TIGIT signalling. Grb2 and β-arrestin 2 were both shown to recruit SHIP-1 to TIGIT complexes to mediate inhibitory signals. Jurkat cells do not express SHIP-1[60], and thus cannot be responsible for the inhibitory signalling observed in

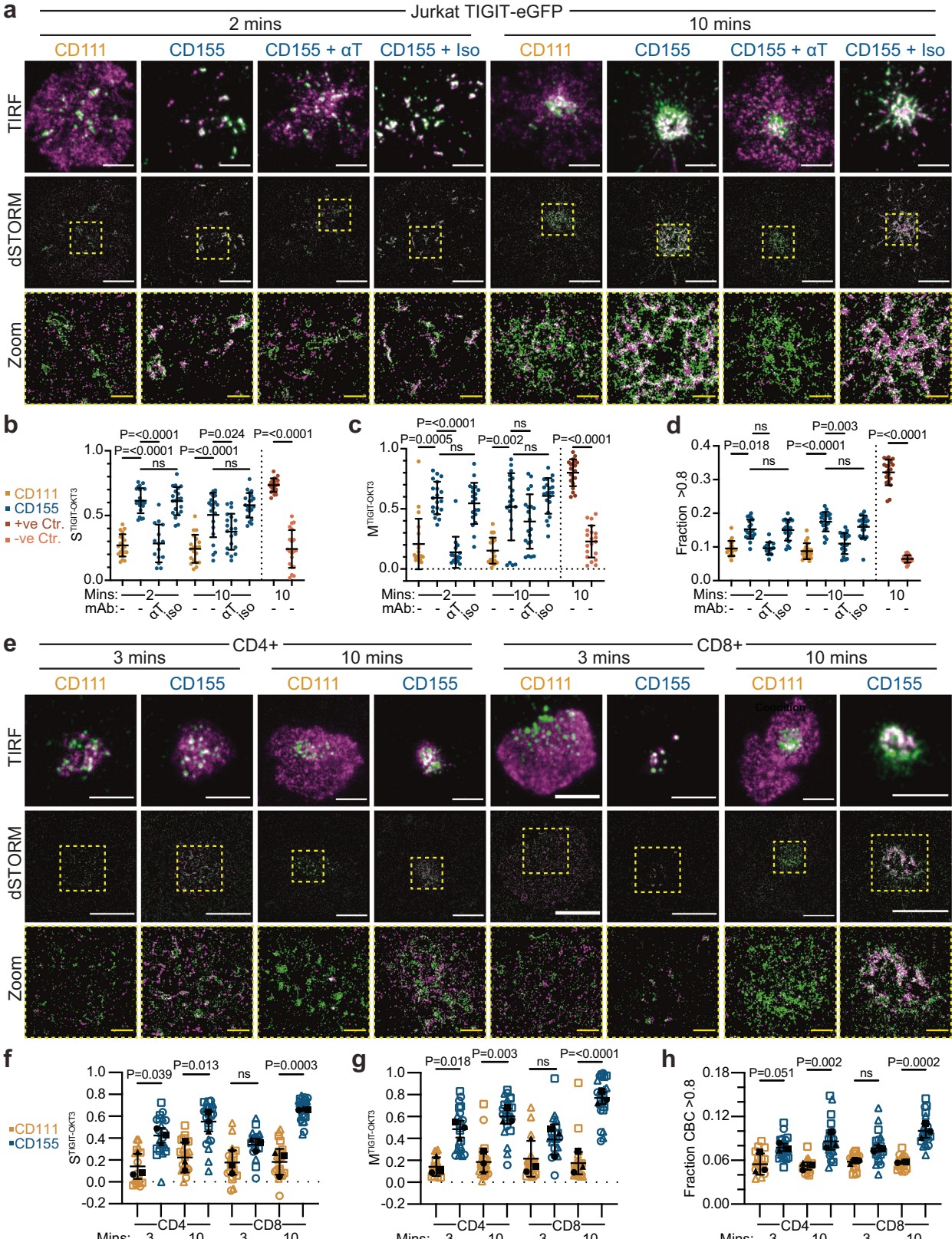

this study. There must, therefore, be other effector molecules that function in Jurkat cells that will likely also function in non-lymphoblastic T cells. We did not observe any TIGIT-mediated reduction in MAPK, AKT or NFκB signalling, and thus the signalling elicited by the ITT-like domain in Jurkat is unclear.

Overall, these data can help inform translation of TIGIT blockade. Combinations of checkpoint inhibitors could synergise to stimulate specific activating pathways and if the mechanisms of TIGIT inhibition were fully understood, patient lymphocytes could be screened to predict responsiveness to TIGIT intervention.

**Fig. 6 | Super resolution microscopy reveals the nanoscale proximity of TIGIT and TCR clusters. a** Two-colour dSTORM imaging of TIGIT-GFP (magenta) in Jurkat cells that have interacted with PLBs loaded with ICAM-1 (100 molecules/μm²), the indicated nectin ligands (400 molecules/μm²) and directly labelled, mono-biotinylated stimulatory TCR antibody OKT3 (green) for the indicated timepoints. Cells pre-incubated with an antagonistic TIGIT antibody (αT) or an isotype-matched control (iso) are shown, as indicated. Representative TIRF and dSTORM images are shown in the top and middle rows, respectively. **b–d** Quantitative analysis of the colocalisation between TIGIT and TCR in Jurkat cells as shown in **a**; $n = \geq 15$ cells per condition, representative of 3 independent experiments, with adjusted $P$ values from a one-way ANOVA with Tukey's multiple comparisons (**b**) or Kruskal-Wallis tests with Dunn's multiple comparisons (**c, d**) are shown. Positive controls represent TIGIT labelled with both a GFP nanobody and a TIGIT antibody, with the negative control representing the same data but with the XY coordinates of one channel being swapped (Supplementary Fig. 12). Mean (±S.D.) Spearman rank correlations (**b**) and Mander's coefficients (**c**) for TIGIT-OKT3 localisations

across each cell. **d** Mean (±S.D.) fraction of localisations that have a score of >0.8 from a coordinate-based colocalisation analysis across single cells. **e** Two-colour dSTORM imaging of TIGIT (magenta) and PLB-bound OKT3 (green) in primary peripheral blood-isolated CD4+ and CD8 + T cells on PLBs, as in **a**. Representative TIRF and dSTORM images are shown in the top and middle rows, respectively. **f–h** Quantitative analysis of the colocalisation between TIGIT and TCR in primary T cells as shown in **e**; $n = \geq 15$ cells (depicted in blue or orange) from 3 individual donors (as shown in black and matched by shape) with adjusted $P$ values from one-way ANOVA with Holm-Šídák's multiple corrections on the means of each donor displayed. Mean (±S.D.) Spearman rank correlations (**f**) and Mander's coefficients (**g**) for TIGIT-OKT3 localisations across each cell. **h** Mean (±S.D.) fraction of localisations that have a score of >0.8 from a coordinate-based colocalisation analysis across single cells. In **a** and **e**, white scale bars = 5 μm, zoomed regions (5 μm x 5 μm; dashed yellow boxes) are displayed below with yellow scale bars = 1 μm. Source data are provided as a Source Data file.

## Methods

### Ethics statement
Peripheral blood was acquired from the National Health Service Blood Transfusion Service under ethics license REC 05/Q0401/108 (University of Manchester, UK) as approved by the West Midlands - Black Country Research Ethics Committee. Tumour samples and associated reference peripheral blood samples were acquired from the commercial vendor Discovery Life Sciences under Task Order DISLS06 as approved by GSK HBSM Due Diligence.

### Cell lines
The Jurkat cell line was obtained from the ATCC (Clone E6-1; ATCC® TIB-152™; RRID:CVCL_0367) and was maintained in RPMI-1640 Medium (R0883; Sigma Aldrich) supplemented with 2 mM L-glutamine (ThermoFisher Scientific), 10% heat-inactivated fetal bovine serum (FBS; Sigma Aldrich) and 50 U/mL Penicillin-Streptomycin (Thermo Fisher Scientific). The Raji cell line was obtained from the ATCC (ATCC® CCL-86; RRID:CVCL_0511) and was maintained as per Jurkat cells. HEK293T cells were obtained from the ATCC (HEK 293T/17; ATCC® CRL-11268; RRID:CVCL_1926) and maintained in DMEM, High Glucose (41965039; Life Technologies) supplemented with 10% heat-inactivated FBS and 50 U/mL Penicillin-Streptomycin (Thermo Fisher Scientific). All cell lines were maintained in 37 °C, 5% CO₂ tissue culture incubators.

### Primary cell isolation
PBMCs were isolated from leucocyte cones of healthy adult donors by density centrifugation using a Ficoll gradient. The PBMC layer was carefully removed and washed to remove platelets. CD4+ and CD8 + T cells were subsequently isolated by negative selection using either a CD4 + T cell isolation kit (130-096-533; Miltenyi Biotec) or a CD8 + T cell isolation kit (130-096-495; Miltenyi Biotec). Cells were maintained in RPMI-1640 Medium supplemented as per Jurkat cells. Primary cells were maintained in 37 °C, 5% CO₂ tissue culture incubators. To stimulate T cells to induce receptor expression, 48-well plates were coated with poly-L-lysine (0.01%; P8920; Sigma-Aldrich) for 10 mins, washed 3 times with ddH₂O, and then coated with 5 ug/mL OKT3 (produced in house; RRID:AB_467057) and 1 ug/mL CD28.2 (16-0289-81; RRID:AB_468926; Thermo Fisher Scientific) in DPBS (D8537; Sigma-Aldrich) for 1 h, washed 3 times in DPBS before cells were plated at $1 \times 10^6$/mL and supplemented with 200 U/mL interleukin-2 (Roche). Cells were used on day 3.

### Mass cytometry analysis of blood and tumour samples
The CyTOF panel included both commercially available and custom-conjugated metal-tagged antibodies which were each validated in-house using a variety of cell types (PBMC, DTC, and cell lines) and culture conditions (Supplementary Table 2). We determined standard

palladium barcoding was not suitable for DTC samples, and instead chose to use live CD45 barcoding solely for the purpose of multiplexing samples during acquisition.

Frozen single cell suspensions of blood and tumour cells were thawed in a 37 °C water bath, slowly added into pre-warmed AIM-V media (Gibco), washed and resuspended in 1 mL AIM-V prior to filtering through a FlowMi® 70uM Cell Strainer (Sigma Aldrich). Cells were plated in a 96-well round-bottom polystyrene plate (Costar) at 2–4 × 10⁶ cells per well. Cells were first stained with 0.25 μM Cisplatin-198Pt in Maxpar® PBS (Fluidigm) for 5 mins, washed, and resuspended in Maxpar® Cell Staining Buffer (Fluidigm) containing Human TruStain FcX™ (BioLegend) for 10 mins. Next the cells were stained with a unique CD45 antibody and the primary surface antibody cocktail for 20 mins. The final surface stain was performed with the secondary anti-FITC antibody under the same conditions. Cells were then fixed and permeabilized using eBioscience™ Foxp3/Transcription Factor Staining Buffer Set (Thermo Fisher Scientific) for 45 mins at 4 °C. After permeabilization, cells were incubated for 10 mins with Human TruStain FcX™ prior to staining with the intracellular antibody cocktail for 30 mins. Finally, cells were resuspended in Cell-ID™ Intercalator-Ir (Fluidigm) in Maxpar® Fix/Perm Buffer (Fluidigm). The 96-well plates were then sealed and stored at 4 °C for 24–72 h prior to acquisition on a CyTOF2 mass cytometer (Fluidigm) equipped with a Super Sampler (Victorian Airship and Scientific Apparatus, LLC).

On the day of sample acquisition, cells were washed twice in Maxpar® Cell Staining Buffer, followed by two washes in Maxpar® Water (Fluidigm). Barcoded samples were then pooled, counted, filtered through a 70um FlowMi® 70uM Cell Strainer, and resuspended to 400,000 cells/mL in water containing 1:10 dilution of EQ Four Element Calibration Beads (Fluidigm). Samples were kept on ice until acquisition, for no longer than 4 h. After acquisition, all files were randomized, normalized, and concatenated using the Fluidigm CyTOF software (Version 6.7.1014) prior to analysis. All biaxial gating and high-dimensional data analysis (UMAP [v3.1], FlowSOM [v3.0.18]) was performed in FlowJo™ v10.7.1 (BD Life Sciences).

Additionally, previously published considerations on how to correctly set gates using mass cytometry[61], which includes accounting for sources of background (abundance sensitivity, isotope purity, and oxide formation) were used to ensure accurate gating. Gates were set based on MMM controls ('metal minus many', similar to FMO in flow), whereby healthy donor PBMCs (same donor each time) were thawed and stained with the full antibody panel and an MMM panel which excluded the CD226 axis antibodies.

### Molecular cloning and plasmid generation
The Gateway™ entry clone for TIGIT (pENTR/Zeo-TIGIT-NoSTOP) was generated by cloning an amplified PCR product (without its STOP codon) from cDNA derived from RNA isolated from peripheral blood

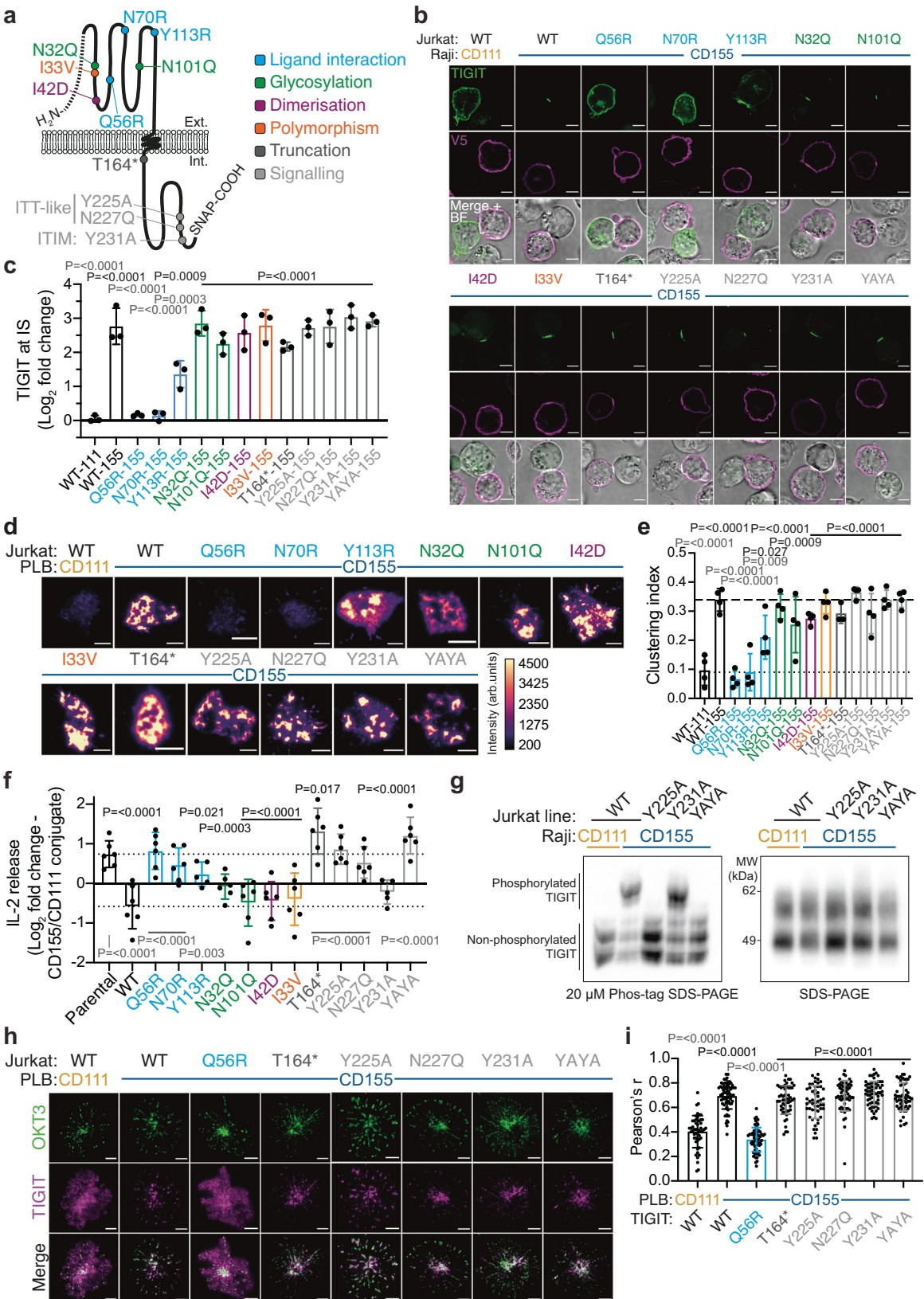

CD8 + T cells stimulated in vitro for 3 days with plate bound OKT3 and CD28.2 (as above), into the donor vector pDONR™/Zeo (12535035; Thermo Fisher Scientific), using BP clonase (11789020; Thermo Fisher Scientific). Primers sequences were: ggggacaagtttgtacaaaaaagcaggc ttcaccatgcgctggtgtctcctc (forward) and ggggaccactttgtacaagaaagctg ggtgaccagtctctgtgaagaagctgca (reverse) and PCR was performed with

Phusion HF DNA polymerase (M0530; New England Biolabs), according to the manufacturer's instructions. pENTR/Zeo-TIGIT-NoSTOP was then used to create the subsequent lentiviral Gateway™ expression clone (pLNT-UbC-TIGIT-eGFP) by recombination with the destination vector pLNT-UbC-#-eGFP[62], with LR clonase (11791020; Thermo Fisher Scientific). TIGIT-SNAP was generated through another LR clonase

**Fig. 7 | ITT-like motif is fundamental to T-cell intrinsic TIGIT-mediated inhibition although functionally null-mutants can still co-cluster with the TCR.**
**a** Schematic depicting individual point mutations introduced into TIGIT-SNAP. **b** Representative confocal microscopy images showing WT and mutant forms of TIGIT-SNAP (green) on the surface of Jurkat T cells conjugated for 20 mins with different Raji B cell populations (either CD111- or CD155-expressing; stained via V5 and shown in magenta). A merged fluorescence-BF image is also provided. **c** Mean log$_2$ fold change (±S.D., $n = 3$ independent experiments) in synaptic TIGIT enrichment in Jurkat T cells, from the conjugates shown in **b**. Adjusted $P$ values from a one-way ANOVA with Šídák's multiple comparisons are displayed, with differences from the WT-111 condition displayed in black and the WT-155 condition displayed in grey. **d** Representative TIRF microscopy images of WT and mutant forms of TIGIT-SNAP at the IS of Jurkat cells that have interacted with PLBs loaded with ICAM-1, and CD111 or CD155 for 20 mins, as in Fig. 2b. Intensities have been scaled equally, and colour scales provided. **e** Mean degree of TIGIT clustering measured from the images shown in **d** (±S.D., $n = 3$–4 independent experiments, as indicated). Adjusted $P$ values from a one-way ANOVA with Dunnett's multiple comparisons are displayed, and coloured as in **c**. **f** ELISA data showing the relative amount of IL-2

released from either parental or different forms of TIGIT-SNAP-expressing Jurkat cells after co-incubation with SEE-pulsed Raji cells. Data is shown as the mean log$_2$ fold changes between Raji-CD155 conjugates compared to Raji-CD111 conjugates, ± S.D. ($n = \geq5$ independent experiments with adjusted $P$ values from a one-way mixed-effects analysis with a Dunnett's multiple comparison test displayed). Differences from the parental condition are displayed above in black and from the WT condition displayed below in grey. **g** Western blot analysis of TIGIT using either Phos-tag SDS-PAGE (left) or standard SDS-PAGE to examine TIGIT phosphorylation in Raji-Jurkat conjugates, as labelled above. Data are representative of 3 independent experiments. **h** Representative TIRF microscopy images of different forms of TIGIT (SNAP labelled; magenta) and the TCR (OKT3 in PLB; green) in Jurkat cells upon interaction with PLBs containing ICAM-1, either CD111 or CD155, and fluorescently labelled OKT3 (100 molecules/µm$^2$), for 10 mins. **i** Mean Pearson's correlation coefficient (±S.D; $n = \geq50$ cells from 2 independent experiments) between TIGIT and OKT3 from the images shown in **h**. Adjusted $P$ values from a Kruskal-Wallis test with Dunn's multiple comparisons are shown, and coloured as in **c**. All scale bars = 5 µm. Source data are provided as a Source Data file.

reaction between pENTR/Zeo-TIGIT-NoSTOP and another destination vector pLX302-SNAP. The pLX302-SNAP vector was constructed in house using pLX302 (Addgene plasmid #25896; gift from David Root) as the backbone. Briefly, the SNAP tag was cloned by PCR from a pSNAP-ADRβ2 control plasmid (Addgene Plasmid #101123; gift from New England Biolabs & Ana Egana), together with a fragment generated from pLX304 (Addgene Plasmid #25890; gift from David Root that excised the V5 fragment) and combined into pLX302 with HIFI DNA assembly (New England Biolabs). Gateway™ cloning with pLX302-SNAP yields a protein with a C-terminal SNAP-tag fusion. Entry clones for the nectin ligands CD111 and CD155 were generated by cloning PCR amplifications (without their STOP codons) from cDNA ORF clones (HG11611-M & HG10109-UT; Sino Biological) into the pDONR/Zeo vector (generating pENTR/Zeo-CD111-NoSTOP and pENTR/Zeo-CD155-NoSTOP, respectively). Primers were ggggacaagtttgtacaaaaaagcagg cttaatggctcggatggggc (CD111 forward), ggggaccactttgtacaagaaagctg ggttcacgtaccactccttcttggaaatga (CD111 reverse), Ggggacaagtttgtacaa aaaagcaggcttaatggcccgagccatgg (CD155 forward) and ggggaccactttg tacaagaaagctgggtgccttgtgccctctgtctgtg (CD155 reverse). Destination clones were then generated by subsequent Gateway™ cloning into pLX302, generating pLX302-CD111 and pLX302-CD155.

## Site-directed mutagenesis
Specific TIGIT mutants were generated in the entry vector pENTR/Zeo-TIGIT-NoSTOP using the Phusion Site-Directed Mutagenesis Kit (F541; Thermo Fisher Scientific), as per manufacturer's instructions. The primers used were as follows: TAGAAACAACGGGG**CAG**ATTTCT GCAGAGA (N32Q_F), TTGTGCCTGTCATCATTCCTGAGGCGAGGG (N32Q_R), AGTCGCTGACCGTG**CAG**GATACAGGGGAGT (N101Q_F), GGAGGGTGAGGCCCAGGCCGGGACCTGGGG (N101Q_R), GAAACAAC GGGGAAC**GTT**TCTGCAGAGAAA (I33V_F), TATTGTGCCTGTCATCATT CCTGAGGCGAG (I33V_R), AAGGTGGCTCTATC**GAC**TTACAATGTCACC (I42D_F), TCTCTGCAGAAATGTTCCCCGTTGTTTCTA (I42D_R), GCATC TATCACACC**CGC**CCTGATGGGACGT (Y113R_F), AGAAGTACTCCCCTG TATCGTTCAC**GGT**CA (Y113R_R), CGGCACAAGTGACC**CGG**GTCAACT GGGAGC (Q56R_F), TGGTGGAGGAGAGGTGACATT**GT**AAGATGA (Q56R_R), TCTGGCCATTTGT**AGA**GCTGACTTGGGGTG (N70R_F), AGCTGGTCCTGCTGCTCCCAGTT**GAC**CTGG (N70R_R), CGAGCTGCA TGAC**GCC**TTCAATGTCCTGAG (Y225A_F), GCACAGTCCTCTCCCCGC TGCTCTCCACAG (Y225A_R), TCAATGTCCTGAGT**GCC**AGAAGCTG GGTA (Y231A_F), AGTAGTCATGCAGCTCGGCACAGTCCTCTC (Y231A_R), CATGACTACTTC**CAG**GTCCTGAGTTACAGA (N227Q_F), CAGCTCGGCACAGTCCTCTCCCCGCTGCTC (N227Q_R), CACCCAGC TTTCTTGTACAAAGTTGGCATT (T164Del_F) and AGTCAACGCGAC CACCACGATGACTGCTGT (T164Del_R). All variants were verified by sequencing using the following primers: AATGTCACCTCTCCTCCACC

(TIGITseq1), GGTGGAGGAGAGGTGACATT (TIGITseqR1), GGAGAG GACTGTGCCGAG (TIGITseqF2), and CTCGGCACAGTCCTCTCC (TIGITseqR2).

## Lentiviral production, transduction, and selection of stable cell lines
All lentivirus plasmids used in this manuscript were transfected into HEK293T cells to produce viral particles containing genes of interest. Briefly, $5 \times 10^5$ HEK293T cells were seeded onto Poly-L-lysine (as above) coated wells of a 6-well plate. 24 h later, 3 µg of total DNA was diluted in serum-free DMEM (volume is 10% of final transfection volume) at a ratio of transgene: viral packaging (psPAX2, Addgene plasmid #12260): viral envelope (pMD2.G, Addgene plasmid #12259) constructs of 9:9:1 (w/w/w). PEI MAX 40 K (24765; Polysciences Catalog), was added at 3x the concentration of DNA (9 µg) and incubated at RT for 15 mins before being added to HEK293T cells. Supernatants were replenished 24 h later with 3 mL of collection media (DMEM + 10% FBS and 1% BSA), and the supernatants containing virus were harvested 24 h later, for two consecutive days. Supernatants were then passed through 0.45 µm pore size cellulose acetate filters (E4780-1453; Starlab), to remove cell debris, and added to a 15 mL falcon tube containing $5 \times 10^4$ cells to be transduced. Polybrene (107689; Sigma-Aldrich) was added to the media (final concentration 8 µg/mL) and centrifuged at 1000 g at 32 °C for 1 h. Cells were washed and resuspended in normal growth media for continued culture. pLX302-transduced cells were selected with 5 µg/mL puromycin (P9620; Sigma-Aldrich) and assessed by flow cytometry. TIGIT-eGFP cells were sorted by fluorescence-activated cell sorting.

## Antibodies
The antagonistic TIGIT antibody used for TIGIT-CD155 blocking experiments was added to cells, on ice, for 10 mins prior to use at 5 µg/mL in imaging assays and 22.5 µg/mL in the 6-h cytokine secretion assays (Clone VSIG9.01; produced by the Center for proteomics, Faculty of Medicine, University of Rijeka). Antibodies used in the CyTOF panel are listed in Supplementary Table 2. For flow cytometry the following antibodies were used: αTIGIT (Clones MBSA43 [#16-9500-82, Thermo Fisher Scientific] and A15153G [372702, BioLegend]), αCD111 (Clone R1.302; 340404; BioLegend), αCD155 (Clone SKII.4; 337622; BioLegend) and αDNAM-1 (Clone DX11; #MA5-28150; Invitrogen). For isotype controls, the clone MOPC-21 (400166; BioLegend) was used for mouse IgG1 κ antibodies and the clone MOPC-173 (400264; BioLegend) for mouse IgG2a κ antibodies. For immunofluorescence experiments the following antibodies were used: αTIGIT (MBSA43; 2.5 µg/mL), GFP-Booster nanobodies (Atto488-, Alexa Fluor 488- and Alexa Fluor 647-conjugated; gba488, gb2AF488 and

gb2AF647; ChromoTek; 1 μg/mL), αDNAM-1 (Clone DX11; 2.5 μg/mL), αCD96 (Clone NK92.39; 5 μg/mL), αV5 (Rabbit polyclonal; NB600-381; Novus Biologicals; 1 μg/mL), αCD19 (Clone HIB19; 302250; BioLegend; 2 μg/mL), αCD4 (Clone MT310; sc-19641; Santa Cruz Biotechnology; 2 μg/mL) and αCD8 (Clone RPA-T8; 301062; 2.5 μg/mL). For WB TIGIT (E5Y1W; 1:2000; Cell Signaling Technology), Beta-Actin-HRP (GTX109639; 1:10000; GeneTex), CD3ζ (6B10.2; 1:1000; sc-1239; Santa Cruz Biotechnology), pCD3ζ (Y142; 1:1000; K25-407.69; #558402; BD Biosciences), Zap70 (D1C10E; 1:1000; #3165; Cell Signaling Technology), pZap70 (Y319; 1:1000; #2701; Cell Signaling Technology), LAT (E3U6J; 1:1000; #45533; Cell Signaling Technology), pLAT (Y220; 1:1000; #3584; Cell Signaling Technology), ERK1/2 (#9102; 1:1000; Cell Signaling Technology), pERK1/2 (E10; 1:2000; #9106; Cell Signaling Technology), AKT (C67E7; 1:1000; #5373; Cell Signaling Technology), pAKT (S473; E4U3U; 1:1000; #23430; Cell Signaling Technology), IκBα (L35A5; 1:1000; #4814; Cell Signaling Technology), pIκBα (S32; 14D4; 1:1000; #2859; Cell Signaling Technology), αRabbit IgG-HRP (#7074; Cell Signaling Technology) and αMouse IgG-HRP (#7076; Cell Signaling Technology) antibodies were used. OKT3 was monobiotinylated according to the protocol previously published[63] using EZ-link™ Sulfo-NHS-LC-LC-biotin (21338; Thermo Scientific; Biotin reagent solution used at 0.1 μg/mL with the antibody at 1 mg/mL in DPBS). Antibodies that were not conjugated directly from the purchaser were then conjugated in house with NHS-Esters of each dye (Alexa Fluor 488 and Alexa Fluor 647; A20000 and A20006; both from Invitrogen; Atto488; 41698; Sigma) and desalted with Zeba spin columns (7 K MWCO; 89882; Thermo Scientific).

## Flow cytometry

To evaluate the abundance of cell surface proteins, living cells were washed in DPBS, and stained with a LIVE/DEAD dye (Zombie, 1:1000 dilution; BioLegend) for 20 min at 4 °C. After subsequent DPBS washing, cells were then fixed with 4% paraformaldehyde at 37 °C for 15 mins. Fixed cells were then washed in DPBS before being blocked in DPBS containing 3% Bovine serum Albumin and 1% Human serum for 20 mins at RT. Cells were then stained in blocking buffer with the indicated primary antibodies for 30 mins at RT. Where secondary antibodies were needed, unbound primary antibodies were washed off with 3 DPBS washes followed by another 30 mins incubation of the secondary antibodies in blocking buffer. Cytoplasmic stains were performed, as above, but with the inclusion of a 5 mins 0.1% Triton-X100 permeabilisation step following fixation. Flow cytometric analysis was carried out on either a BD Fortessa X20 or a BD FACSymphony (Becton Dickinson), and all analysis was carried out with the FlowJo software (v10.4.2; FlowJo).

## Sample preparation for microscopy – conjugate imaging

Cell conjugates were formed by incubating T cell and Raji cells together in eppendorf tubes (1:1 ratio) before centrifugation at 50 g for 30 s. Cells were then incubated for 2 mins at 37 °C in a heat bath, before being carefully resuspended with a wide-bore 1000 μL tip and plated onto 0.01% PLL-coated Lab-Tek II imaging chambers (Nunc), and placed into 37 °C, 5% CO2 tissue culture incubators. At the indicated timepoints, cells were fixed with the addition of paraformaldehyde to a final concentration of 4% and incubated at 37 °C for 15 mins. Cells were then washed in DPBS and where permeabilisation was required, this was performed with 0.1% Triton-X100 for 5 mins before blocking in DPBS containing 3% Bovine serum Albumin and 1% Human serum for either 20 mins at RT, or overnight at 4 °C. Cells were labelled with antibodies in blocking buffer for 1 h at RT, and where secondary antibodies were used washed with DPBS three times before being labelled for 1 h in blocking buffer. Samples were washed and kept in DPBS for imaging unless otherwise stated.

## Sample preparation for microscopy – planar lipid bilayers

Cells were plated onto PLB-coated Lab-Tek II imaging chambers (Nunc), and all experiments were performed at 37 °C, 5% CO2 in tissue culture incubators. Cells were fixed with the addition of paraformaldehyde to a final concentration of 4% and incubated at 37 °C for 15 mins. Cells were then washed in DPBS with the serial addition and removal of 500 μL DPBS (5 time; 200 μL of liquid was always maintained within the well to prevent desiccation of the PLB). Cells were permeabilised, blocked and labelled as above. For live cell imaging, cells were resuspended in growth media, and added to wells containing PLBs that contained an equal volume of DPBS. Imaging proceeded at 37 °C, 5% CO2. For single-colour STORM imaging, slides were immersed in 0.22 μm-filtered STORM imaging buffer (560 μg/mL glucose oxidase, 34 μg/mL catalase, 1% β-mercaptoethanol, 25 mM glucose, 5% glycerol (all from Sigma-Aldrich), and 25 mM HEPES/DPBS pH 8 (Thermo Fisher Scientific), refreshed regularly to maintain a low-oxygen environment for optimal fluorophore blinking. For two-colour STORM imaging, slides were immersed in 0.22 μm-filtered OxEA imaging buffer (50 mM β-MercaptoEthylAmine hydrochloride (MEA, Sigma-Aldrich), 3% (v/v) OxyFluor™ (Oxyrase Inc.), 20% (v/v) sodium DL-lactate solution (Sigma-Aldrich) in DPBS, pH adjusted to 8–8.5 with NaOH), as previously published[64].

## Preparation of supported and planar lipid bilayers

Liposomes were prepared as previously described in detail[63]. Briefly, liposomes were generated by extrusion with both the lipids and the extruder obtained from Avanti Polar Lipids, Inc. Bilayers were made up of lipid compositions, containing different percentages of 4 mM stock solutions of DOPC (1,2-dioleoyl-sn-glycero-3-phosphocholine), Ni-NTA (1,2-dioleoyl-sn-glycero-3-[(N-(5-amino-1-carboxypentyl)iminodiacetic acid)succinyl]) and Cap-Biotin (1,2-dioleoyl-sn-glycero-3-phosphoethanolamine-N-(CAP biotinyl)).

To create bilayers on silica beads (generating BSLBs), 5 μm silica beads (SS05N; Bangs Laboratories, Inc) were washed 3 times with ddH2O and then DPBS before being incubated with liposome solutions containing 87.5% DOPC, 12.5% Ni-NTA, at a ratio of 2:1 (liposome:bead; v/v). These were then vortexed at maximum speed for 30 seconds. Blocking solution (1% BSA, 100 μM nickel sulphate in DPBS) was added and vortexed again for 10 seconds, and incubated for 20 mins at RT. Following blocking, ligands could be added by incubating recombinant proteins in DPBS with BSLBs. BSLBs were then washed 3 times with 1% BSA/DPBS and are re-suspended in media containing cells at the required E:T ratio (1:2.5; Jurkat:aAPCs). As His-ligands were used, this was performed in serum free, or Ni-NTA depleted FBS.

To quantify ligand loading, bead suspensions were then aliquotted to individual wells of a V-bottom 96-well plate, so that each well contained 5 × 10^5 beads. Supernatants were removed by centrifugation at 600 g for 2 mins, being careful not to remove all the liquid. Proteins were added at varying concentrations to ensure equal densities (i.e. for each nectin ligand ~14 nM incubated for 1 h at 37 °C achieved 400 molecules/μm²). Protein densities were measured on beads using Quantum MESF beads (Bangs Laboratories, Inc), as per manufacturer's instructions for each batch of lipids and proteins used.

Planar lipid bilayers were generated on 8-well Lab-Tek II chamber slides (Nunc). All wells of the slides to be used were first washed with 1 M HCl in 70% ethanol, for 30 mins at RT, and washed 5 times with sterile ddH2O. Following this, wells were washed with 0.45 μm filtered 10 M NaOH for 15 mins at RT, and washed 5 times with sterile ddH2O, followed by drying under an argon stream. 200 μL liposome solution (87.5%:12.5% DOPC:Ni-NTA when using His-ligands only and 87.3%:12.5%:0.2% DOPC:Ni-NTA:Cap-Biotin when using bilayers with biotinylated ligands) were added to each well. These were washed by sequential addition and removal of 500 μL DPBS, 5 times to remove excess lipid. 1% BSA with 100 μM nickel sulphate was added to block. If

biotinylated ligands were used, 200 μL of blocking solution was removed and 1 μg/mL Streptavidin in DPBS was added and incubated for 20 mins. Streptavidin conjugated to Alexa-Fluor 647 was used to test the mobility of PLBs and BSLBs, by FRAP. Excess streptavidin is removed by 5 sequential DPBS washes and then ligand is loaded into the PLB, at the desired concentration. After the final incubation, 5 more DPBS washes were performed and then samples were equilibrated with multiple washes in the buffer to be used for cellular incubation.

## Jurkat-Raji Staphylococcal Enterotoxin E (SEE) assay
Cells were counted and washed in serum-free media. 30 mins prior to conjugation, SEE (ET404; Toxin Technologies) was added to Raji cells (30 ng/mL for experiments with TIGIT mutants in Fig. 7 and 60 ng/mL for experiments shown in Fig. 2h, Supplementary Fig. 8a and Supplementary Fig. 16) in serum free media and incubated for 30 mins. Excess SEE was washed with normal growth media before cells were resuspended in normal growth media for use in the assay. Meanwhile, Jurkat cells were counted and mixed with Raji cells (At a ratio of Jurkat:Raji of 2:1 for cytokine release assays and 3:1 for WB analysis). For cytokine release assays cells were then spun at $50\,g$ for 1 min to bring cells into proximity and placed at 37 °C, 5% $CO_2$ for 6 h before supernatants were collected following centrifugation at 500 g for 5 mins to remove cells. For WB analysis both Raji and Jurkat cells were washed once with serum-free RPMI and then serum starved for 2 h prior to conjugation. SEE was added to Raji cells halfway through serum starvation and excess washed off with serum-free RPMI. Immediately prior to cell mixing, both cell lines were incubated on ice for 10 mins and then $5 \times 10^6$ Jurkat cells were mixed with $1.66 \times 10^6$ Raji cells in a final volume of 400 μL, on ice. Cells were then spun at 50 g for 1 min to bring cells into proximity, at 4 C, before being placed in a 37 C water bath to initiate cell signalling. Zero timepoints were not added to the water bath and immediately washed and lysed. At the indicated timepoints, cells were taken from the water bath and placed on ice, 1 mL ice cold PBS was added and then centrifuged at 500 g for 5 min at 4 C. PBS was removed and 200 μL RIPA buffer (50 mM Tris-HCl pH 7.4, 150 mM NaCl, 1% NP-40, 0.5% Deoxycholate, 0.1% Sodium dodecyl sulfate, 1X Halt Protease and Phosphatase inhibitor cocktail (78441; Thermo Fisher Scientific)) was added to lyse cells, on ice.

## Enzyme-linked immunosorbent assay (ELISA)
IL-2 was measured in cell supernatants using an ELISA. Nunc Max-iSorp™ 96-well plates (44-2404-21; Thermo Fisher Scientific) were coated with 1 μg/mL human capture IL-2 antibody (555051; BD Pharmingen; clone 5344.111) in 50 mM carbonate bicarbonate (C3041; Sigma-Aldrich), overnight at 4 C. Excess antibody was removed with 3 washes with washing buffer (DPBS with 0.05% Tween20). Wells were blocked in blocking buffer (1% BSA in DPBS, 0.22 μm filtered), for 1 h at RT. Samples and standards (rhIL-2; 554603; BD Biosciences) were incubated with the capture antibody for 2 h at RT, before being washed 3 times with wash buffer and subsequent incubation with 1 μg/mL detection antibody (555040; BD Pharmingen; clone B33-2). Following 3 more washes, wells were incubated with streptavidin-HRP (554066; BD Biosciences) at RT for 30 mins, and washed 3 more times. TMB solution (MP Bio) was then added to each well, the reaction proceeded for 10 mins before being stopped with 0.5 M $H_2SO_4$, and optical density at 450 nm detected on a spectrophotometer.

## Microscopy: image acquisition
Confocal microscopy images were acquired through a 100X oil immersion objective (NA 1.40) on a laser scanning confocal microscope (TCS SP8 STED CW; Leica Microsystems). Excitation was performed by sequential combination using a pulsed white-light laser and emission was detected using time-gated (from 0.8 to 6.0 ns) HyD hybrid photon detectors operating in standard mode. Images were acquired by LASAF software (v3.3) and exported for processing and

analysis as raw data. 3D-TauSTED was performed on the same microscope as for confocal imaging. STED images were acquired with the FLIM mode active, using 3 line accumulations and 3 frame averages. The scan speed was set to 200 Hz, zoom set to 2.5, frame size set to 1024 × 512 pixels, resulting in a pixel size of 45 nm and a pixel dwell time of 1.2 μs. The pinhole was set to 0.96 AU at 580 nm, and 70 z-planes were recorded with a step size of 120 nm. Excitation lasers (WLL) were set to 5% at 518 nm and 10% at 554 nm with depletion of both with 5% 660 nm STED laser. TIRF microscopy was performed on an inverted microscope (Leica SR 3D-GSD) fitted with a HC PL APO 160X oil immersion lens (NA 1.43) and an EMCCD camera (Andor iXon Ultra 897), and the TIRF laser penetration depth set to 150 nm for all wavelengths used. Single colour STORM images were acquired by excitation with the 647-nm laser (15%), acquiring 15,000 frames with a 11-ms exposure time and an electron multiplier gain set to 120. 2 colour STORM imaging was performed sequentially, imaging 647 first followed by 488. Excitation with the 647-nm laser was performed at 15% power for 7500 frames with the addition of a 15% 405-nm laser after the first 1000 frames. This was then followed immediately by excitation with the 488-nm laser (at 50% power) for 7500 frames with the addition of a 15% 405-nm laser after the first 1000 frames.

## Fluorescence recovery after photobleaching (FRAP)
FRAP was performed on a laser scanning confocal microscope (TCS SP8 STED CW; Leica Microsystems) to i) assess the mobility of ligands in lipid bilayers and ii) to visualise the recovery of TIGIT and CD155 molecules within clusters. Mobility of ligands in PLBs or BSLBs was assessed using Streptavidin or ligand directly conjugated to Alexa-Fluor 647. Briefly, 1 μg/mL of either protein was loaded into bilayers, washed as described above and imaged using the 100× oil objective (NA, 1.4). 5 μm × 5 μm regions were bleached with 100% 660 nm STED laser and imaged every second for 60–90 s with the WLL set at 650 nm. Mobility could then be assessed by plotting the recovery within the bleached region over time, compared to adjacent 5 μm × 5 μm regions that had not been bleached. We routinely observe 100% recovery with $T_{1/2} \sim 5$ s. To assess TIGIT and CD155 clusters in cells on PLBs, the same setup was used, except that TIGIT-GFP was imaged and bleached using the WLL at 488 nm, and CD155-AF647 imaged and bleached with the WLL at 650 nm, in defined ROIs that covered the entirety of a visible cluster. Recovery was then assessed by normalising recovery to clusters of similar intensity that were not bleached.

## Western Blotting
Cells were lysed in RIPA buffer, protein concentration quantified and normalised using a BCA assay (23227; Thermo Fisher Scientific), reduced in 1X NuPAGE™ LDS Sample Buffer (NP0007; Thermo Fisher Scientific) supplemented with 2 mM DL-Dithiothreitol (D9779; Sigma-Aldrich) at 70 °C for 10 mins and separated in NuPAGE™ 4 to 12%, Bis-Tris gels (NP0323; Thermo Fisher Scientific) in 1X NuPAGE™ MOPS SDS Running Buffer (NP0001; Thermo Fisher Scientific), according to manufacturer's instructions. Separated proteins were transferred onto 0.45 μm PVDF membranes, using the Trans-Blot Turbo transfer system (Bio-Rad). Membranes were blocked in 5% Milk (For non phospho-antibodies; 70166; Merck Millipore) or 5% BSA (For phospho-antibodies; A9647; Sigma-Aldrich) in Tris-buffered saline with 0.1% Tween-20 (TBST) for 1 h at RT and incubated with primary antibody in either 1% Milk or 1% BSA in 0.1% TBST overnight at 4 °C. Membranes were then washed 5 times in TBST before being probed with secondary antibodies in block buffer for 1 h at RT. Membranes were then washed again 5 times in TBST before being revealed with Clarity™ Western ECL Substrate (#1705060; Bio-Rad) and imaged using the ChemiDoc MP imaging system (Bio-Rad).

Phos-tag SDS PAGE was carried out using custom-made gels, according to the manufacturer's instructions. Briefly, 20 μM Phos-tag affinity reagent AAL-107 (#300-93523; Wako Chemicals) was

incorporated into 6% polyacrylamide gels, using $MnCl_2$. The same lysates were used to perform standard SDS-PAGE, to show as controls.

## Microscopy: image analysis
To calculate the clustering index, we initially selected a region of interest around cells in TIRF images and extracted all pixel intensities with a custom macro in Fiji. Mean intensities were then calculated for each cell to account for cell-to-cell heterogeneous expression levels and a threshold of 1.5× the calculated means was set to determine the clustering index. For each cell, the fraction of pixels above the threshold was determined as the clustering index. Mean and fraction counting was performed in R (v3.5.3; R Core Team, 2019). Clustering analysis of single molecule localisation data was performed using custom MATLAB scripts available at Github (https://github.com/quokka79/RegionFinder; https://github.com/quokka79/ClusterFields). Data were also processed in Microsoft Excel (2016).

## TauSTED processing
STED images were processed within the LAS X software (v3.5.7), using the fluorescence lifetimes to filter pixels from the image. Tau background suppression was activated, with Tau strength set to 200 and denoise to 50. A time gate was also applied to exclude photons outside of a 0.4–6 ns window.

## Single-molecule localization microscopy data
STORM images were reconstructed with the ThunderSTORM software[65]. Raw images were filtered to remove noise and enhance blinking events from single fluorophores (wavelet filter B-spine method, order 3, scale 2). Initial event detection was determined using the local maximum localization method, with a threshold for the peak intensity set to 2 times the standard deviation of the F1 wavelet, and 8-neighborhood connectivity for each pixel screened. Subpixel event localization was calculated using an integrated Gaussian point-spread function and maximum likelihood estimator with a fitting radius of 5 pixels and an initial sigma of 1.6 pixels. STORM events were then filtered so that only events with the following criteria were retained: intensity >600 photons, sigma (range between 50 and 200) and uncertainty <30 nm. This filtered data set was then corrected for sample drift using cross-correlation of images from 3 bins at a magnification scale of 5. Re-blinking events were minimised by merging filtered, and drift-corrected events that were localised within 50 nm and 20 frames of the initial detection. The threshold for merging was determined by evaluating the fluorophores on each antibody at a very low density on a glass surface[66]. In this manner, the number of localised events should be close to the number of fluorophores imaged. MATLAB (R2018a, MathWorks) was used for the clustering analysis of the STORM data, with all data being calculated and plotted within the software.

## Statistics & Reproducibility
Statistical analysis and graphical data representation in the manuscript was performed with Prism (v8.4.2, GraphPad), and statistical tests used are indicated within figure legends. Normality of data was also assessed in Prism to inform on the appropriate tests to be used. The exact data that is plotted within graphs (i.e., mean, SD or SEM) is also indicated in the figure legends and in the text. No statistical method was used to predetermine sample size, and data exclusions have been denoted within figure legends or within the methods. Experiments were neither randomised, nor investigators blinded to allocation during experiments and outcome assessment.

## Reporting summary
Further information on research design is available in the Nature Portfolio Reporting Summary linked to this article.

## Data availability
Data supporting the findings of this study are included within the Article and its supplementary figures, tables. Videos analysed for Fig. 3f and generated for a more in-depth visualisation of Supplementary Fig. 9 are included as Supplementary Videos 1–2. Uncropped immunoblots used in the manuscript are provided in the Supplementary Information. Source data are provided with this paper. All data can be made available upon request. Cell lines and plasmids generated for use in this study can also be made available upon request. Source data are provided with this paper.

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

## Acknowledgements

The authors thank Gareth Howell and the Manchester Collaborative Centre for Inflammation Research (MCCIR) Flow Cytometry Facility for cell sorting, Andy Shepherd (GSK) for producing the OKT3 antibody, Kevin Stacey and Alicia Evans for primary cell isolations, the Genome Editing Unit for the generation of the pLX302-SNAP construct and all members of the Davis laboratory for helpful discussions. This work was supported through funding from GSK, the Medical Research Council (MR/W031698/1 to D.M.D), the Wellcome Trust (Investigator Award; 110091/Z/15/Z to D.M.D.) and the Manchester Collaborative Centre for Inflammation Research (funded by a precompetitive open-innovation award from GSK, AstraZeneca, and The University of Manchester, United Kingdom).

## Author contributions

Conceptualization, J.D. Worboys and D.M.D.; Data curation, J.D. Worboys; Formal Analysis, J.D.Worboys, K.N.V., R.K.H., A.A. and M.A.C.; Funding acquisition, D.M.D.; Investigation, J.D.Worboys, K.N.V., M.A.C., F.P.P., J.G.-M., W.H.Z. and R.K.H.; Methodology, J.D.Worboys, K.N.V., A.A., K.S.H. and R.K.H. Project administration, G.M.T.; Resources, S.J. and T.L.-R.; Supervision, G.D.S.C.D.M., J.D. Waight and D.M.D.; Visualization, J.D.Worboys; Writing – original draft, J.D. Worboys; Writing – review & editing, J.D. Worboys, R.K.H., M.A.C., S.J., T.L.-R., A.A., K.L.J., C.R., J.D. Waight and D.M.D.

## Competing interests

F.P.P., G.D.S.C.D.M., G.M.T., J.D.Waight, K.N.V. and M.A.C. are employed by GSK and DMD is a consultant and advisor to GSK. The remaining authors declare no competing interests.
