## [Peer Review File · Nature Communications]

TIGIT can inhibit T cell activation via ligation-induced nanoclusters, independent of CD226 co-stimulationREVIEWER COMMENTS

Reviewer #1 (Remarks to the Author):

In the present study, Worboys et al. explore mechanisms by which the co-inhibitory receptor TIGIT limits human T cell function. Several studies have demonstrated that TIGIT can impair T cell activation by preventing binding of a shared ligand (CD155) to the costimulatory receptor CD226, with one recent study suggesting these cell extrinsic interactions to be TIGIT's primary mechanism of inhibition. However, the authors observe that a minority of T cells from patients with cancer co-express TIGIT and CD226 and therefore reason that the inhibitory effects of TIGIT in patients with cancer must predominantly be independent of CD226. While this conclusion perhaps ignores significant previous publications in humans indicating that in other tumors, TIGIT and CD226 are co-expressed (Lucca LE, et al *Neurol Neuroimmunol Neuroinflamm.* 2020) and the possibility that TIGIT+CD226- populations may still limit the availability of CD155 in the tumor microenvironment, thereby decreasing co-stimulatory signals for CD226+ population. Moreover, the authors ignored previous investigations showing the CD155 engagement of TIGIT on human Tregs in CD226 knockdown still exhibit changes in function (Lucca LE et al. *JCI Insight.* 2019, Lozano E, et al *J Immunol.* 2012)

Nevertheless, the authors provide an elegant illustration of how CD155 binding induces a dynamic TIGIT reorganization within the immune synapse to then generate cell-intrinsic inhibitory signals. Using previously annotated TIGIT mutants, the authors demonstrate that TIGIT nanocluster formation is mostly dependent on CD155 binding whereas cell-intrinsic inhibitory effects are primarily dependent on cytoplasmic signaling domains. We do note that the precise downstream signaling mechanisms remain unexplored.

1. Can the authors comment on how the observed frequency of CD226+/TIGIT+ observed compares with prior reports (i.e. Banta et al., Lucca LE, et al) where >10% of CD8+ T cells co-expressed CD226 and TIGIT)?
2. Can the authors also demonstrate the relationship between CD226 expression and CD155 in the tumor microenvironment (similar to in Figure 1H)? Are the correlations comparable to what has previously been described?
3. Did the authors observe differences in TIGIT/CD226 expression between tumor types or other clinicopathologic variables?
4. In Figure 2H, can the authors comment on why anti-TIGIT only partially rescues IL-2 secretion in TIGIT-SNAP Jurkat cells? Were any dose titration studies performed?
5. In Figures 3D-E, can the authors provide baseline flow staining of TIGIT and CD226 for the primary CD4 and CD8 T cells used? How might CD226 contribute to the differences in clustering pattern observed?
6. The authors demonstrate how CD155 ligation induces TIGIT to form nanoclusters around the TCR. Likewise, the same occurs when CD155 ligates CD226. Are the authors able to comment on how CD226 and TIGIT clusters interact upon CD155 ligation?
7. Are the authors able to perform phosphorylation studies to more precisely confirm the functional effects of the mutations in the ITT-like and ITIM domains?
8. Line 286: "laterally" should be changed to "lateral"
9. Line 384: "CD11" should be corrected to "CD111"

Reviewer #2 (Remarks to the Author):

Inhibitory receptors play a key role in limiting T cell activity, in particular in guarding against autoimmune disease. Blocking them is of therapeutic interest in enhancing the anti-tumor immune response. In their mechanism of action, it is of interest whether they can trigger inhibitory signaling or compete with costimulatory receptors for ligand access. Both mechanisms of action are potential enhanced by molecular proximity to stimulatory receptors. Here Worboys et al investigate the inhibitory receptor TIGIT in this context. Using a 40 parameter flow cytometry panel, the authors show that TIGIT expression is high in tumor-infiltrating T cells, most

prominently on regulatory T cells, commonly in parallel with its dominant ligand CD155, rarely in parallel with the costimulatory receptor CD226 that shares CD155 as its ligand. This expression pattern suggests an autonomous inhibitory role of the TIGIT/CD155 interaction. In the remainder of the manuscript Worboys et al. investigate the localization of TIGIT at immune interfaces and its ability to regulate secretion of the T cell cytokine IL-2. TIGIT accumulates at the interface between Jurkat T cells and primary CD4+ and CD8+ T cells with Raji B cells. The presence of TIGIT and CD155 inhibits IL-2 secretion. By substituting the Raji B cell with a supported lipid bilayer, the authors facilitate use of higher resolution microscopy methods including the single molecule localization microscopy approach dSTORM. Thus, they show that TIGIT clusters dynamically and that TIGIT is enriched in the molecular proximity of the T cell receptor. Using a series of TIGIT mutants, Worboys et al. show that ligand binding is required for TIGIT clustering but signaling motifs in the cytoplasmic domain for inhibition of IL-2 secretion. Together these data suggest that TIGIT may function by generating an inhibitory signal in the direct molecular proximity of the T cell receptor. This is a substantial contribution to our understanding of TIGIT. The technical execution of the manuscript is highly rigorous, and data are carefully interpreted.

Some minor additions to the discussion would be welcome.

1 For technical reasons all clustering data are generated in the interaction of T cells with supported lipid bilayers. Membrane topology and cytoskeletal transport processes differ at the physiological more relevant interaction of T cells with antigen presenting cells. It remains unclear how the bilayer data translate into this setting. This caveat should be mentioned.

2 The importance of TIGIT clustering for its inhibitory function remains unresolved. This would require a TIGIT mutant that can't cluster but can still signal. The TIGIT mutants used that don't cluster don't do so because they can't engage CD155. Hence, they don't signal either. The TIGIT mutant that lacks the entire cytoplasmic domain still clusters but enhances IL-2 secretion. This suggests that some element of TIGIT clustering actually counteracts its inhibitory function. The caveat of the uncertain role of TIGIT clustering should be discussed.

3 The authors suggest that blocking TIGIT on regulatory T cells where it is highly expressed will reduce their immunosuppressive capacity (row 566). At first sight, blocking an inhibitory receptor with intrinsic inhibitory signaling capability on an immunosuppressive cell type should enhance their function, hence increase suppression. This should be clarified.

4 Typo row 384, should be CD111 instead of CD11.

POINT-BY-POINT RESPONSE TO REVIEWER COMMENTS

Reviewer #1 (Remarks to the Author):

In the present study, Worboys et al. explore mechanisms by which the co-inhibitory receptor TIGIT limits human T cell function. Several studies have demonstrated that TIGIT can impair T cell activation by preventing binding of a shared ligand (CD155) to the costimulatory receptor CD226, with one recent study suggesting these cell extrinsic interactions to be TIGIT's primary mechanism of inhibition. However, the authors observe that a minority of T cells from patients with cancer co-express TIGIT and CD226 and therefore reason that the inhibitory effects of TIGIT in patients with cancer must predominantly be independent of CD226. While this conclusion perhaps ignores significant previous publications in humans indicating that in other tumors, TIGIT and CD226 are co-expressed (Lucca LE, et al *Neurol Neuroimmunol Neuroinflamm.* 2020) and the possibility that TIGIT+CD226- populations may still limit the availability of CD155 in the tumor microenvironment, thereby decreasing co-stimulatory signals for CD226+ population. Moreover, the authors ignored previous investigations showing the CD155 engagement of TIGIT on human Tregs in CD226 knockdown still exhibit changes in function (Lucca LE et al. *JCI Insight.* 2019, Lozano E, et al *J Immunol.* 2012)

Firstly, we would like to thank both reviewers for taking the time to read and review our manuscript.

We thank the reviewer for drawing our attention to these important gaps in our introduction and discussion. We agree that different tumours may very well present environments in which TIGIT and CD226 expression varies. To address this, we have now modified the text as follows:

- In the abstract (line 45) '*Given that most TIGIT-expressing cells within tumours do not co-express CD226, this is likely its dominant mechanism of action.*' becomes '*Within the subset of tumours where TIGIT-expressing cells do not commonly co-express CD226, this will likely be the dominant mechanism of action.*'
- Line 93. In the introduction we have added a phrase to read '*In this study we demonstrate that T and NK cells within both the blood and tumours of renal and lung cancer patients seldom co-express TIGIT and CD226, suggesting that within certain tumours TIGIT acts predominantly independently of CD226 cis interactions.*'
- Line 107. We have changed the title of the first results section to read '*TIGIT and CD226 co-expression is infrequent across T and NK cell subsets in renal and lung cancer patient tumours*'.
- Line 577. We have also changed the text within the discussion to now read '*Here, we show that TIGIT and CD226 are infrequently co-expressed on both peripheral blood and particularly so on tumour infiltrating T lymphocytes in renal and lung cancer*'.
- Line 584. We have again clarified '*The data we present would suggest that this model of TIGIT inhibition is likely to contribute to a small fraction of the immune cells in particular cancers. This may not be the case for all tumours, as TIGIT and CD226 were co-expressed in glioblastoma multiforme and in melanoma (Lucca et al, 2020 & Fourcade et al, 2018).*'

To acknowledge the previously published finding that TIGIT and CD226 can compete to function within Tregs we have now included the following sentence in the discussion (line 587): '*In the case of co-expression, TIGIT and CD226 may indeed compete to signal, as has been shown in Treg cells (Lozano et al, 2012).*'

To acknowledge the work in Treg cells demonstrating TIGIT signalling we have also included a reference to Lozano et al (2012) at line 71 in the introduction: '*TIGIT possesses T-cell intrinsic inhibitory potential*'.

We also agree that TIGIT expression on CD226⁻ cells could limit ligand availability within the tumour microenvironment and would thus also contribute to TIGIT-mediated inhibition. Indeed, we cannot rule out the effects of TIGIT⁺CD226⁻ populations on CD226⁺ populations, and thus we have amended the manuscript to specifically raise that possibility:

- Moreover, we have modified the text in the discussion (line 589) to now read '*Where TIGIT is singly expressed, our data imply that TIGIT signalling itself, is important for its inhibitory function and would include T-cell intrinsic inhibition, TIGIT-mediated CD155 signalling on APCs⁴, and cell extrinsic inhibition through reduced ligand availability for CD226 signalling.*'
- We have also included the following at line 620 '*Upon TIGIT blockade, this population may be boosted by costimulatory signals that are more likely to occur through CD226-CD155 interactions.*'

Nevertheless, the authors provide an elegant illustration of how CD155 binding induces a dynamic TIGIT reorganization within the immune synapse to then generate cell-intrinsic inhibitory signals. Using previously annotated TIGIT mutants, the authors demonstrate that TIGIT nanocluster formation is mostly dependent on CD155 binding whereas cell-intrinsic inhibitory effects are primarily dependent on cytoplasmic signaling domains. We do note that the precise downstream signaling mechanisms remain unexplored.

We are delighted to see the reviewer describe our manuscript as elegant. We have endeavoured to explore the downstream signalling mechanisms, details below.

1. Can the authors comment on how the observed frequency of CD226⁺/TIGIT⁺ observed compares with prior reports (i.e. Banta et al., Lucca LE, et al) where >10% of CD8⁺ T cells co-expressed CD226 and TIGIT)?

In Banta *et al*, Figure 3 shows co-expression of various receptors with CD226 and/or CD28. They find that of total CD8⁺ T cells, ~10% are CD28⁺/CD226⁺ and just under 20% are CD28⁻/CD226⁺. Within each subset they then present the total percentage that also co-express TIGIT (~40% in both cases), which they say is a significant number. They do not show the number of TIGIT-expressing cells within the total CD8 population, and they do not show the UMAPs (Uniform Manifold Approximation and Projections) for TIGIT. This would equate to 40% of 30% of all CD8 T cells, equal to ~12%. We report this number to be ~5% in tumours in our manuscript. Note that in our manuscript the MMI (Mean Metal Intensity) that sets the threshold for a positive population was 10, whereas the average TIGIT MMI reported in the Banta *et al*. publication on CD226⁺ cells was ~3, which suggests very low level TIGIT expression. For direct comparison Banta *et al*. report MMIs for PD-1 ~20-40, TIM-3 ~2-10, CD27 ~15-75 and CD103 ~10-160 within the same subsets. As the authors do not show their biaxial plots it is hard to look at the relative expression between CD226⁺/TIGIT⁺ cells. Additionally, we have used previously published considerations on how to correctly set gates using mass cytometry (Nicholas KJ *et al* Cytometry A; 2016), which includes accounting for sources of background (abundance sensitivity, isotope purity, and oxide formation). We set our gates based on MMM controls ('metal minus many', similar to FMO in flow) – in every experiment we thawed and stained a healthy donor PBMC (same donor each time) with the full antibody panel and an MMM panel which excluded the CD226 axis antibodies to ensure we could set the gates properly. This information has been added to the methods section to be clear (lines 757-763).

In Lucca *et al*, figure 3 reports that ~18% of CD8+ cells co-express CD226 and TIGIT, and ~14% of CD4+ T cells, yet these values are taken from example biaxial plots. The authors only present double positive cells within a CD226+ gate (20% for CD4 and 75% for CD8) and therefore it is difficult to make a fair comparison of the data. They also show that TIGIT and CD226 levels remain the same between blood and TIL but TIGIT increases and CD226 decreases within GBM TILs. TIGIT+ within CD226+ subsets remain unchanged in both subsets.

2. Can the authors also demonstrate the relationship between CD226 expression and CD155 in the tumor microenvironment (similar to in Figure 1H)? Are the correlations comparable to what has previously been described?

We thank the reviewer for the suggestion and have now included this data in the manuscript. We do not observe CD226 on any T cell subset to positively correlate with CD155 expression on myeloid APCs (now figure 1h, and the graph showing the %PD-1+ on T cell vs CD155 on mAPC has been subsequently moved to Extended Data Figure 2d), nor on NK cells (CD226 incorporated into the graph in Extended Data Figure 3h).

To our knowledge, no study has looked for a correlation between myeloid-expressed CD155 and T cells or NK cells in the TME. *In vitro* experiments have demonstrated CD155's ability to downregulate CD226 on NK cells (Carlsten *et al*, *J Immunol* 2009; Chauvin *et al*, *Clin Cancer Res* 2020) and CD8+ T cells (Braun *et al*, *Immunity* 2020). In the latter publication they demonstrate using IHC an inverse relationship between CD226 on CD8+ T cells and CD155 expression within the TME in melanoma samples. Although the source of CD155 in the Braun publication is not limited to mAPCs and thus difficult to compare directly, we do see similar overall trends.

We have edited the text to reflect the newly included data at line 184.

3. Did the authors observe differences in TIGIT/CD226 expression between tumor types or other clinicopathologic variables?

Upon the reviewer's request, we reanalysed the data to look at expression between different cell subsets in the different tumours analysed (see graphs below). As can be seen we do not observe much difference between cells from different tumour origins, with only ccRCC CD8s being statistically significant from NSCLC CD8s. We do not include these data in the manuscript as it does not add.

4. In Figure 2H, can the authors comment on why anti-TIGIT only partially rescues IL-2 secretion in TIGIT-SNAP Jurkat cells? Were any dose titration studies performed?

We thank the reviewer for noticing this. Originally, we used an equivalent concentration of antibody in Jurkat-Raji co-culture assays as we had used to observe complete blockade of the clustering and synaptic enrichment in imaging assays but had never titrated the antibody specifically within the co-culture model. We agree that this is important due to the observed partial rescue and thus we have now performed a dose titration to optimise TIGIT inhibition with the mAb used in Raji-Jurkat co-culture assays. The graph below shows IL-2 secretion from Jurkat TIGIT-SNAP cells co-cultured with SEE-pulsed Raji CD155 cells (E:T at 2:1), incubated with increasing concentrations of our anti-TIGIT mAb.

The data included in the original submission used the TIGIT mAb at 33 nM (5 μ g/mL), which is clearly sub-optimal. Thus, we repeated the experiment that comprised Figure 2H and Extended Data Figure 7A using the TIGIT mAb and isotype control at 150 nM, which improved the observed rescue. These new graphs now directly replace those in Figure 2h and Extended Data Figure 7a (now Extended data figure 8).

Note that in the original submission we had used 30 ng/mL SEE to pulse label Raji cells in this assay. However, we now use 60 ng/mL as that results in more IL-2 production that has proved to be more robust. This has resulted in slightly lower co-stimulatory effects of the CD155-DNAM interaction in the Raji-Jurkat assay but greater reduction in IL-2 release caused by TIGIT-CD155. We have amended the methods sections to accurately reflect the different antibody and superantigen concentrations used (lines 987-989).

5. In Figures 3D-E, can the authors provide baseline flow staining of TIGIT and CD226 for the primary CD4 and CD8 T cells used? How might CD226 contribute to the differences in clustering pattern observed?

As per the reviewer's suggestion, we now provide baseline flow staining of the primary cells used in the assay. We use primary T cells that were stimulated for 3 days with anti-CD3 and anti-CD28 mAbs and IL-2, to boost TIGIT expression. As can be seen in the plots, consistent with previous publications, TIGIT expression increases with activation in both subsets. Additionally, 3-day activation increases the number of TIGIT⁺CD226⁺ cells permitting a subsequent new analysis of TIGIT and CD226 as suggested by the reviewer (see below). This data forms part of a new figure (Extended data figure 6).

It is an interesting idea that CD226 may alter TIGIT clustering. To test this, we turned to stimulated primary T cells that have populations of both TIGIT⁺CD226⁻ and TIGIT⁺CD226⁺ cells. Thus, we incubated primary CD4⁺ and CD8⁺ T cells with stimulatory PLBs containing ICAM-1, anti-CD3 mAb (OKT3) and CD155 for 10 mins and labelled the cells for TIGIT with AF647 and CD226 with Atto488 to differentiate TIGIT⁺CD226⁻ and TIGIT⁺CD226⁺ cells. We then imaged a total of 120 cells from 3 independent blood

donors by TIRF microscopy and performed single colour dSTORM of TIGIT to enable detailed clustering analysis (Extended data figure 14a). TIGIT expression was not significantly different between subsets (Extended data figure 14b and c). Clustering analysis of TIRF images showed no differences in TIGIT clustering in any subset (Extended data figure 14d). Furthermore, detailed clustering analysis of single molecule localisation data also reveal similar protein synaptic density, cluster sizes and TIGIT densities within clusters (Extended data figure 14e-h). Thus, CD226 does not seem to impact TIGIT clustering. These data now form a new figure in the manuscript (Extended data figure 14).

6. The authors demonstrate how CD155 ligation induces TIGIT to form nanoclusters around the TCR. Likewise, the same occurs when CD155 ligates CD226. Are the authors able to comment on how CD226 and TIGIT clusters interact upon CD155 ligation?

Given that we did not observe a high frequency of cells co-expressing TIGIT and CD226 from tumours, we did not ask this question in our initial submission. However, to address this now, we incubated activated CD4+ and CD8+ T cells with PLBs containing ICAM-1, OKT3 and either CD111 or CD155 and performed 2-colour STORM microscopy of TIGIT and CD226. These new data form a new figure (Extended data figure 13).

TIGIT and CD226 do indeed converge upon CD155 ligation within dense clusters (Extended data figure 13a-d). Thus, TIGIT and CD226 signalling may compete locally to inhibit or stimulate T cells. We now explicitly state this in the discussion (Line 588).

Interestingly, we observe little colocalisation in the unligated condition, which could be expected given the hypothesis that TIGIT and CD226 interact *in cis*. It is possible the interaction is transient, or that TIGIT-CD226 *cis* interactions block the epitope of either antibody. Thus, although we don't observe TIGIT-CD226 *cis* interactions, we cannot rule out the possibility that they occur on primary cells and have added this to the discussion (Lines 601-605).

7. Are the authors able to perform phosphorylation studies to more precisely confirm the functional effects of the mutations in the ITT-like and ITIM domains?

We agree that this is important. To address this, we set out to confirm the phosphorylation of TIGIT through Western blotting techniques. Initially we had attempted to immunoprecipitate TIGIT through the SNAP fusion protein (using the SNAP-trap beads from Chromotek), and although we were always successfully able to IP TIGIT we consistently failed to detect any signal using a phosphotyrosine antibody cocktail (4G10 platinum; successfully used in previous TIGIT publications). However, we could confirm TIGIT phosphorylation using a Phos-tag gel. Phos-tag gels incorporate functional molecules that bind phosphorylated ions into the polyacrylamide gel so that protein migration is affected by phosphorylation levels. Using Phos-tag we were successfully able to show two important observations: i) that TIGIT is phosphorylated in a ligand dependant manner and ii) that Y225 is the major site of phosphorylation (see below). This data now forms a panel of figure 7 (panel g).

To understand the downstream signalling consequences of TIGIT signalling, Western blotting was used to look at the dynamic phosphorylation changes in Jurkat-Raji co-cultures. Specifically, parental, TIGIT-SNAP (WT) and TIGIT-SNAP (Y225A/Y231A; YAYA) Jurkat cells were incubated with SEE-pulsed Raji CD155 cells for 0, 3, 10 or 30 minutes before being lysed and subject to SDS-PAGE and Western blotting analysis (Extended data figure 16a). As TIGIT localises to the TCR upon co-ligation, we looked at the phosphorylation of CD3 ζ (Y142), ZAP70 (Y319) and LAT (Y220) and found no reduction in signalling in TIGIT-expressing Jurkat cells. Additionally, we looked at the phosphorylation of ERK1 and ERK2 (T202/Y204), AKT (S473) and I κ B α (S32) as these have all been described to be reduced by TIGIT in

previous publications. Again, we found no significant reductions in phosphorylation in TIGIT-expressing Jurkats. We did observe a mild reduction in phosphorylation of I κ B α but no degradation of the total form (Extended data figure 16a-b). As Jurkat cells do not express SHIP-1, this may explain why we do not observe a reduced phosphorylation of MAPK, AKT and NF κ B signalling, as all have been shown to be SHIP-1 dependent. Thus, together with all other observations in the manuscript, we show that TIGIT signalling is vital for its inhibitory function and can signal through signalling pathways currently unknown. These data are presented in a new figure (Extended data figure 16) and discussed (Lines 655-656 and 678-680).

8. Line 286: “laterally” should be changed to “lateral”

We thank the reviewer for spotting this typo, which has now been corrected.

9. Line 384: “CD11” should be corrected to “CD111”

We thank the reviewer for spotting this typo, which has also now been corrected.

Reviewer #2 (Remarks to the Author):

Inhibitory receptors play a key role in limiting T cell activity, in particular in guarding against autoimmune disease. Blocking them is of therapeutic interest in enhancing the anti-tumor immune response. In their mechanism of action, it is of interest whether they can trigger inhibitory signaling or compete with costimulatory receptors for ligand access. Both mechanisms of action are potential enhanced by molecular proximity to stimulatory receptors. Here Worboys et al investigate the inhibitory receptor TIGIT in this context. Using a 40 parameter flow cytometry panel, the authors show that TIGIT expression is high in tumor-infiltrating T cells, most prominently on regulatory T cells, commonly in parallel with its dominant ligand CD155, rarely in parallel with the costimulatory receptor CD226 that shares CD155 as its ligand. This expression pattern suggests an autonomous inhibitory role of the TIGIT/CD155 interaction. In the remainder of the manuscript Worboys et al. investigate the localization of TIGIT at immune interfaces and its ability to regulate secretion of the T cell cytokine IL-2. TIGIT accumulates at the interface between Jurkat T cells and primary CD4+ and CD8+ T cells with Raji B cells. The presence of TIGIT and CD155 inhibits IL-2 secretion. By substituting the Raji B cell with a supported lipid bilayer, the authors facilitate use of higher resolution microscopy methods including the single molecule localization microscopy approach dSTORM. Thus, they show that TIGIT clusters dynamically and that TIGIT is enriched in the molecular proximity of the T cell receptor. Using a series of TIGIT mutants, Worboys et al. show that ligand binding is required for TIGIT clustering but signaling motifs in the cytoplasmic domain for inhibition of IL-2 secretion. Together these data suggest that TIGIT may function by generating an inhibitory signal in the direct molecular proximity of the T cell receptor. This is a substantial contribution to our understanding of TIGIT. The technical execution of the manuscript is highly rigorous, and data are carefully interpreted.

We are delighted with these comments from the reviewer and thank them for the assessment that our manuscript is highly rigorous, and data carefully interpreted.

Some minor additions to the discussion would be welcome. 1 For technical reasons all clustering data are generated in the interaction of T cells with supported lipid bilayers. Membrane topology and cytoskeletal transport processes differ at the physiological more relevant interaction of T cells with antigen presenting cells. It remains unclear how the bilayer data translate into this setting. This caveat should be mentioned.

We agree with the reviewer and have added a comment to the discussion (lines 632-636) *'It is important to note that most clustering data in this manuscript is generated using supported lipid bilayers, that differ from the cell surface of an APC. Thus, there may be effects from other aspects of APCs, such as the sub-synaptic actin, or specific nanoscale organisation of target cell ligands, which are not captured in our experiments.'*

2 The importance of TIGIT clustering for its inhibitory function remains unresolved. This would require a TIGIT mutant that can't cluster but can still signal. The TIGIT mutants used that don't cluster don't do so because they can't engage CD155. Hence, they don't signal either. The TIGIT mutant that lacks the entire cytoplasmic domain still clusters but enhances IL-2 secretion. This suggests that some element of TIGIT clustering actually counteracts its inhibitory function. The caveat of the uncertain role of TIGIT clustering should be discussed.

We agree with the reviewer that we cannot conclude on the importance of clustering for TIGIT's function. We have now explicitly stated this within the discussion (line 647-649) *'However, as all*

mutants that could engage CD155 were able to cluster, we could not directly discriminate the role of clustering for TIGIT function.'

Regarding the cytoplasmic null mutant, we already discuss how clustering may lead to enhanced signalling (lines 640-645):

'Intriguingly, we observed clustering in both the YAYA and T164 mutants that lack inhibitory signalling capacity, and a concomitant increase in IL-2 release from these mutant-expressing Jurkats in SEE-Raji conjugates. Therefore, it is possible that the clustering formed upon TIGIT-CD155 interaction can be stimulatory, possibly through force generation or increased adhesion when no inhibitory signals are present.'*

3 The authors suggest that blocking TIGIT on regulatory T cells where it is highly expressed will reduce their immunosuppressive capacity (row 566). At first sight, blocking an inhibitory receptor with intrinsic inhibitory signaling capability on an immunosuppressive cell type should enhance their function, hence increase suppression. This should be clarified.

We thank the reviewer for drawing our attention to this potential confusion. In effector T cells inhibitory receptors terminate T cell responses, while in Tregs they are thought to promote their suppressor function (Lucca & Dominguez-Villar, 2020). To clarify this, we have added a reference (Kurtulus *et al*, 2015 at line 617) whereby they demonstrate how TIGIT+ Treg cells have more suppressive capacity, and thus TIGIT blockade can prevent TIGIT-driven suppressor function in these populations.

4 Typo row 384, should be CD111 instead of CD11.

We thank the reviewer for spotting this typo, which has now been corrected.

REVIEWERS' COMMENTS

Reviewer #1 (Remarks to the Author):

The authors have addressed the comments and should be accepted for publication

Reviewer #2 (Remarks to the Author):

my minor concerns have been addressed

REVIEWERS' COMMENTS

Reviewer #1 (Remarks to the Author):

The authors have addressed the comments and should be accepted for publication

Reviewer #2 (Remarks to the Author):

my minor concerns have been addressed

We would like to thank both reviewers for taking the time to review and improve our manuscript.